# MULTI-AGENT REINFORCEMENT LEARNING WITH SHARED RESOURCE FOR INVENTORY MANAGEMENT

## ABSTRACT

We consider inventory management (IM) problem for a single store with a large number of SKUs (stock keeping units) in this paper, where we need to make replenishment decisions for each SKU to balance its supply and demand. Each SKU should cooperate with each other to maximize profits, as well as compete for shared resources e.g., warehouse spaces, budget etc. Co-existence of cooperation and competition behaviors makes IM a complicate game, hence IM can be naturally modelled as a multi-agent reinforcement learning (MARL) problem. In IM problem, we find that agents only interact indirectly with each other through some shared resources, e.g., warehouse spaces. To formally model MARL problems with above structure, we propose shared resource stochastic game along with an efficient algorithm to learn policies particularly for a large number of agents. By leveraging shared-resource structure, our method can greatly reduce model complexity and accelerate learning procedure compared with standard MARL algorithms, as shown by extensive experiments.

## 1 INTRODUCTION

Inventory management (IM) problem has long been one of the most important applications in the supply-chain industry (Nahmias & Smith, 1993). Its main purpose is to maintain a balance between the supply and demand of stock keeping units (SKUs) in a supply chain by optimizing replenishment decisions of each SKU. Besides leading to profit increment and operational cost reduction, efficient IM can even give rise to better services to customers. However, it is quite a challenging task in practice, especially when there are lots of SKUs involved in the supply-chain. Particularly, while all SKUs should cooperate with each other to achieve high profits, they also need to compete for shared resources e.g., warehouse spaces, budget etc. Such co-existence of cooperation and competition renders IM a complicated game that is hard to address.

Traditional methods usually reduce IM problems to solving dynamic programming problems. However, these approaches often rely on some unrealistic assumptions such as i.i.d. demand, deterministic leading time, etc. Moreover, as the state space grows exponentially along with some key factors like leading time and number of SKUs (Gijsbrechts et al., 2019), the corresponding dynamic programming problems become intractable due to the curse of dimensionality. Because of these limitations, many approaches based on approximate dynamic programming are proposed to solve IM problems in different settings (Halman et al., 2009; Fang et al., 2013; Chen & Yang, 2019). While these approaches perform well in certain scenarios, they heavily rely on problem-specific expertise or assumptions e.g., the zero or one period leading time assumption in (Halman et al., 2009), hence can hardly generalize to other settings. In contrast, reinforcement learning (RL) based methods, with fast inference speed, can generalize to various scenarios in a data-driven manner. However, it is usually too costly to train a global policy that can make decisions for all SKUs, since the training efficiency can be notably curtailed because of the large global state and action space (Jiang & Agarwal, 2018)). To further address the training efficiency issue, it is natural to adopt the multi-agent reinforcement learning (MARL) paradigm, where each SKU is regarded as an agent whose state and action spaces are localized and only contain information relevant to itself.

There are currently two popular paradigms to train MARL in the literature: independent learning (Tan, 1993) and joint action learning (Lowe et al., 2017). Despite of their success in many scenarios, these two MARL paradigms also exhibit certain weaknesses that restrain their effectiveness

in solving IM problems. On one hand, if applying independent learning, policy training of one agent simply treats all other agents as parts of the stochastic environment, hence is hard to converge due to non-stationary of the environment. On the other hand, joint action learning usually learns a centralized critic conditioned on the joint action and state spaces of all agents, which can easily become intractable with increasing number of SKUs. Furthermore, it could be quite time-consuming to sample data from joint simulator for a great number of SKUs, since it usually involves much computation on many internal variables caused by complex agent interactions.

To address these challenges, we took a closer look at the IM problem and find that there exists a special structure that can be leveraged to design more effective MARL paradigm. Particularly, each agent in IM only interacts with others through the shared resource, e.g., warehouse spaces. We introduce an auxiliary variable to represent the whole inventory level, implying the available shared resource, for all SKUs, and refer to this variable as *context*. From the MARL perspective, one agent can be influenced by other agents only through such context. The context dynamics actually reflect the collective behaviors of all agents. And, conditioned on context dynamics, one agent's state transition and reward function are independent of other agents. In this way, leveraging context as an additional input for each agent's policy/value functions enables us to both avoid the non-stationary problem caused by independent learning and mitigate the intractable centralized critics learning caused by exploding state and action spaces. Moreover, introducing context dynamics inspires us to build a local simulator for each agent in order to facilitate more efficient policy learning for each agent.

Based on this structure with context dynamics, we propose a shared-resource stochastic game to model the IM problem. Specifically, we introduce a surrogate objective function to optimize the return of agent $i$ conditioned on all other agents, denoted by $-i$. Here, we make two approximations to get the surrogate objective function: 1) rearranging the sampling process by first sampling context dynamics then sampling local state/action/reward for each agent $i$; 2) using context dynamics sampled by previous policies. Based on above surrogate objective, our algorithm consists of two iterative learning procedures: 1) obtaining context dynamics from joint simulator, 2) updating policy for each agent by data sampled from its respective local simulator conditioned on collective context dynamics. By decoupling each agent from all others with a separate training procedure, our method can greatly reduce model complexity and accelerate learning procedure, as shown by extensive experiments. It is worthwhile to mention that our method is not limited to IM, but can also be applied to many other applications with shared-resource structure, e.g., portfolio management (Ye et al., 2020), smart grid scheduling (Remani et al., 2019), etc.

Our contributions are summarized as follows:

- We propose shared-resource stochastic game to capture the problem structure in IM, that agents only interact with each other through shared resource.

- We propose a novel algorithm that leverages the shared-resource structure to solve IM problem efficiently.

- Extensive experiments demonstrate that our method outperforms existing MARL algorithms on both profits and computing efficiency.

## 2 BACKGROUND

### 2.1 STOCHASTIC GAMES

We build our work on stochastic games (SGs) (Shapley, 1953), since each SKU in IM problem has its own profit (reward) to optimize. A Stochastic Game is defined by a tuple $\left(\mathcal{N}, \mathcal{S}, \{\mathcal{A}^i\}_{i \in \mathcal{N}}, \mathcal{T}, \{R^i\}_{i \in \mathcal{N}}, \gamma\right)$, where $\mathcal{N} = \{1, \cdots, n\}$ denotes the set of $n > 1$ agents, $\mathcal{S}$ denotes the state space observed by all agents, $\mathcal{A}^i$ denotes the action space of agent $i$. Let $\mathcal{A} := \mathcal{A}^1 \times \cdots \times \mathcal{A}^n$, then $\mathcal{T} : \mathcal{S} \times \mathcal{A} \to \Delta(\mathcal{S})$ denotes the transition probability from any state $s \in \mathcal{S}$ to any state $s' \in \mathcal{S}$ after taking a joint action $a \in \mathcal{A}$; $R^i : \mathcal{S} \times \mathcal{A} \times \mathcal{S} \to \mathbb{R}$ is the reward function that determines the immediate reward received by agent $i$ for a transition from $(s, a)$ to $s'$; $\gamma \in [0, 1)$ is the discount factor. We can formulate the joint policy of other agents'

as $\pi^{-i} = \prod_{j \in -i} \pi^j$. Each agent $i$ optimizes its policy $\pi^i : \mathcal{S} \to \Delta\left(\mathcal{A}^i\right)$ to maximize its own long-term reward, which is conditioned on other agents' behavior, defined as

$$max_{\pi^i} \eta^i(\pi^i, \pi^{-i}) = \mathbb{E}_{(s_t, a_t^i, a_t^{-i}) \sim \mathcal{T}, \pi^i, \pi^{-i}}[\sum_{t=0}^{\infty} \gamma^t r_t^i]. \tag{1}$$

We will illustrate the shared resource structure of IM problem in Section 2.2, which motivates us to propose shared-resource stochastic game as a special case of stochastic game to capture such structure in Section 3.1.

## 2.2 INVENTORY MANAGEMENT WITH SHARED RESOURCE

While a typical setting for IM shall involve a supply network of multi-echelon including stores, warehouses and factories, we simplify the setting to ease our presentation. In the following, we shall focus on scenarios with one store and multiple SKUs. We further assume that there is an upstream warehouse that can fulfill requirements from the store perfectly. Our objective is to learn high-quality replenishing policies for each SKU in the store, particularly when there are a large number of SKUs. As replenishing decisions for SKUs in stores should directly consider consumption behaviors of customers and competitions from other SKUs due to limited resources like spaces, budget etc., they are more challenging to optimize comparing to SKUs in warehouses. It is worthwhile to mention that, due to the flexibility of RL algorithms, our method can also be applied to more complex settings with multi-echelon, fluctuate supply, non-deterministic leading time etc.

Similar to previous work, we follow the multi-agent RL (MARL) framework with decentralized agents, each of which manages inventory of one SKU in the store. We assume that the store has $n$ SKUs in sell, all of which share a common space that can store up to $I_{max}$ units at the same time. Replenishing decisions of each SKU are made on discretized time steps, which are days in our paper.

For each time step $t$ and SKU $i$, let $\dot{I}_t^i \in \mathbb{Z}$ denote units of $i$ that are in stock. Hence, the following constraint shall hold for all time steps:

$$\forall t \geq 0. \sum_{i=1}^{n} \dot{I}_t^i \leq I_{max} \tag{2}$$

At any time step $t$, the agent corresponding to SKU $i$ may place a replenishment order to request $O_t^i \in \mathbb{Z}$ units of products from its upstream warehouse. These replenishment orders cannot be fulfillment instantly, but will take several time steps, referred as leading time, before these products are delivered to the store. Let $L_i$ denote the leading time of SKU $i$ and $T_t^i \in \mathbb{Z}$ its total units in transit at the time step $t$. Meanwhile, demands from customers $D_t^i$ may consume inventory of SKU $i$ and cause an actual sale of units $S_t^i \in \mathbb{Z}$. Due to the possibility of out-of-stock, $S_t^i$ may be less than $D_t^i$. Formally, dynamics of these variables can be described as follows:

$$S_t^i = \min\left(D_t^i, \dot{I}_t^i\right) \tag{3}$$

$$T_{t+1}^i = T_t^i - O_{t-L_i+1}^i + O_{t+1}^i \tag{4}$$

$$\hat{I}_t^i = \dot{I}_t^i - S_t^i + O_{t-L_i+1}^i \tag{5}$$

$$\rho = \begin{cases} 0 & \text{if } \sum_{i=1}^{n} \hat{I}_t^i \leq I_{max} \\ \frac{\sum_{i=1}^{n} \hat{I}_t^i - I_{max}}{\sum_{i=1}^{n} O_{t-L_i+1}^i} & \text{otherwise} \end{cases} \tag{6}$$

$$\dot{I}_{t+1}^i = \dot{I}_t^i - S_t^i + \lfloor(1-\rho)O_{t-L_i+1}^i\rfloor \tag{7}$$

As we mentioned before, due to all SKUs share a common space, it may happen that the storage overflows when ordered products arrive. In this paper, we assume that the excess SKUs will be discarded proportionally according to $\rho$ defined in Eq. 6, which corresponds to the overflowing ratio if we accept all coming SKUs without considering the space constraint. To calculate $\rho$, we also introduce an extra variable $\hat{I}_t^i$, which we call afterstate of $\dot{I}_t^i$. Intuitively, it denotes units of SKU $i$ in stock at the end of the $t$-th time step if we omit the capacity constraint. We shall note that other manners e.g., prioritizing all SKUs, to resolve space overflows are possible. The algorithm that we

will introduce in the following section will also apply to these settings. Undoubtedly, such behaviors will cause extra operational cost which should be avoided as much as possible in our replenishing decisions.

The intermediate profit $Pt_t^i$ of the $i$-th SKU is calculated according to the following equation:

$$Pt_t^i = p_i S_t^i - q_i O_t^i - o\mathbb{I}\left[O_t^i > 0\right] - h\dot{I}_t^i \tag{8}$$

where $p_i$ and $q_i$ are the unit sale price, unit procurement price for the $i$-th SKU, respectively, and $o$ and $h$ are the order cost and unit holding cost, respectively. $\mathbb{I}[\cdot]$ is an indicator function which equals to one when the condition is true and zero otherwise. The order cost reflects the fixed transportation cost or the order processing cost, and yields whenever the order quantity is non-zero.

For convenience, we summarize all notations in Table 2 in Appendix B, where we also give an example with 2 SKUs in Fig. 4 to further illustrate the whole procedure.

## 3 METHODOLOGY

### 3.1 SHARED-RESOURCE STOCHASTIC GAME

In this section, we show that IM problem introduced in Section 2.2 can be formulated as a shared-resource stochastic game, where each agent is only influenced by other agents through a shared resource pool. We define a shared-resource stochastic game as a tuple $\left(\mathcal{N}, \{\mathcal{S}^i\}_{i \in \mathcal{N}}, \mathcal{C}, \{\mathcal{A}^i\}_{i \in \mathcal{N}}, \mathcal{T}, \{R^i\}_{i \in \mathcal{N}}, \gamma\right)$, where $\mathcal{N} = \{1, \cdots, n\}$ denotes the set of $n > 1$ agents, $\mathcal{S}^i$ denotes the state space of agent $i$, $\mathcal{C}$ denotes the *context* space observed by all agents, $\mathcal{A}^i$ denotes the action space of agent $i$. Here, the *context* represents the occupied capacity of shared-resource. Let $\mathcal{S} := \mathcal{S}^1 \times \cdots \times \mathcal{S}^n$ and $\mathcal{A} := \mathcal{A}^1 \times \cdots \times \mathcal{A}^n$, then $\mathcal{T} : \mathcal{S} \times \mathcal{C} \times \mathcal{A} \to \Delta(\mathcal{S} \times \mathcal{C})$ denotes the transition probability, which can be decomposed as follows. The context is affected by all agents, i.e., $c_{t+1} \sim P_c(\cdot \mid c_t, s_t, a_t)$. Since we are dealing with resource, the transition function of context usually has some additive structure with respect to all agents, which we will illustrate later in Section 3.2. Given context $c_{t+1}$, the transition function of state for each agent is independent with other agents, i.e., $s_{t+1}^i \sim P_s^i(\cdot \mid s_t^i, a_t^i, c_{t+1})$. Reward function is also independent with other agents given context, $r_t^i \sim R^i(s_t^i, a_t^i, c_{t+1})$. We refer $P_s^i$ and $R^i$ as *local* transition and reward functions. $\gamma \in [0, 1)$ is the discount factor. Each agent $i$ optimizes its own policy $\pi^i : \mathcal{S}^i \times \mathcal{C} \to \Delta\left(\mathcal{A}^i\right)$ to maximize the expected long-term reward conditioned on other agents' policies $\pi^{-i}$, defined as

$$max_{\pi^i}\eta^i(\pi^i, \pi^{-i}) = \mathbb{E}_{(s_t, c_t, a_t^i, a_t^{-i}) \sim \mathcal{T}, \pi^i, \pi^{-i}}[\sum_{t=0}^{\infty} \gamma^t r_t^i] \tag{9}$$

Given the above definition, an IM problem can be formulated as a shared-resource stochastic game by letting $r_t^i = Pt_t^i$, $a_t^i = O_t^i$, and $c_t = \sum_{i=1}^n \dot{I}_t^i$ for all SKU $i$ and time step $t$. Moreover, state of each SKU $i$ will be a concatenation of information like $T_t^i$, $p_i$, its historical actions and demands etc. A detailed description of the state space can be found in the Appendix E.3.

### 3.2 SURROGATE OBJECTIVE WITH CONTEXT DYNAMICS

We now introduce how to optimize the return for agent $i$ conditioned on other agents' behaviors. Roughly speaking, the context dynamics reflect the collective behaviors of other agents. It is possible to estimate the objective function approximately for each agent $i$ only based on its local transition dynamics, local reward function and the context dynamics. First, we give a detailed description of the transition model of shared resource $c$ from each agent's perspective. Given such dynamics for the shared resource, we then show how to approximate the objective function in Eq. 9 by rearranging the sampling process. Finally, we replace the policies for sampling context dynamics by old policies from the previous iteration to accelerate the decentralized training.

### 3.2.1 CONTEXT DYNAMICS AND LOCAL SIMULATOR

We use $\dot{c}_t^i$ to represent the amount of resource occupied by agent $i$ at time step $t$ and $\dot{c}_t$ the total amount of occupied resource. We further let $\dot{c}_t^{-i}$ denote the amount of resource occupied by all

agents but $i$. Since the capacity has the additive structure, we have the following equations:

$$\dot{c}_t = \sum_i \dot{c}_t^i;$$
$$\dot{c}_t = \dot{c}_t^i + \dot{c}_t^{-i}. \tag{10}$$

Similarly, we denote the afterstate of $\dot{c}_t^i$ as $\hat{c}_t^i$. From the perspective of agent $i$, it can view the context $\dot{c}_t^{-i}$ as a part of the environment. Hence, given $s_t^i$, $\dot{c}_t^i$, and $a_t^i$, $P(\hat{c}_t^i \mid s_t^i, a_t^i, \dot{c}_t^i)$ represents how the replenishment decision influences $\hat{c}_t^i$. Due to the additive structure, we have $\hat{c}_t = \hat{c}_t^i + \hat{c}_t^{-i}$. Then by applying a resource overflow resolution procedure as in Eq. 6, we obtain $\dot{c}_{t+1}^i$ representing state of the resource in the next step.

We use notation $c_t$ to represent $(\hat{c}_{t-1}, \dot{c}_t)$. We refer $(s^i, c^i)$ as the state for agent i, and $c^{-i}$ as the context. For each agent, we have the following sampling process $\mathcal{T}^i$ given the context dynamics of $c^{-i}$,

$$c_{t+1}^i \sim P_c^i(\cdot \mid s_t^i, a_t^i, c_t^i, c_t^{-i})$$
$$s_{t+1}^i \sim P_s^i(\cdot \mid s_t^i, a_t^i, c_{t+1}^i, c_{t+1}^{-i}) \tag{11}$$
$$r_t^i \sim R^i(s_t^i, a_t^i, c_{t+1}^i, c_{t+1}^{-i}).$$

Given the context dynamics for $c^{-i}$, we can build a local simulator for agent $i$ according to the above equations. To ease our presentation, we only consider one kind of shared resource i.e., warehouse spaces in the above formulation. However, it can be extended to support multiple resources as long as these resources have the additive structure as in Eq. 10.

### 3.2.2 SURROGATE LOSS FUNCTION

To leverage local simulators, we rearrange the sampling process for evaluating $\eta(\pi^i, \pi^{-i})$ as follows. First, we follow the regular sampling process in Eq. 9, and get samples of $c_t^{-i}$. Then, we re-sample $(s_t^i, a_t^i, c_t^i)$ based on samples of $c_t^{-i}$ following Eq. 11. It is worth noting that we make an approximation here. Given samples for $c_t^{-i}$, we actually need to inference the posterior of $(s_t^i, a_t^i, c_t^i)$. However, since we consider scenarios with lots of agents, it is reasonable to assume that each agent $i$ has limited influence on its context $c^{-i}$. Therefore, we can assume that $c_t^{-i}$ has limited influence on the inference of $(s_t^i, a_t^i, c_t^i)$ and sample the latter directly according to Eq. 11. By rearranging the sampling process, we obtain the following surrogate objective,

$$max_{\pi^i} \tilde{\eta}^i(\pi^i, \pi^{-i}) = \mathbb{E}_{(c_t^{-i}) \sim \mathcal{T}, \pi^i, \pi^{-i}} \mathbb{E}_{(s_t^i, a_t^i, c_t^i) \sim \mathcal{T}^i, \pi^i} [\sum_{t=0}^{\infty} \gamma^t r_t^i]. \tag{12}$$

In practice, it is quite costly to sample from the joint dynamics $\mathcal{T}$, but much cheaper to sample from the local dynamics $\mathcal{T}^i$. To leverage data samples efficiently, we propose to use the samples from previous policies in our objective function, which is a common trick in reinforcement learning. Specifically, we use samples of $c_t^{-i}$ collected by policies $(\pi_{old}^i, \pi_{old}^{-i})$ of the previous iteration, and further rewrite the surrogate objective function as follows:

$$max_{\pi^i} \hat{\eta}^i(\pi^i, \pi_{old}^i, \pi_{old}^{-i}) = \mathbb{E}_{(c_t^{-i}) \sim \mathcal{T}, \pi_{old}^i, \pi_{old}^{-i}} \mathbb{E}_{(s_t^i, a_t^i, c_t^i) \sim \mathcal{T}^i, \pi^i} [\sum_{t=0}^{\infty} \gamma^t r_t^i]. \tag{13}$$

As long as current policies stay close to previous ones, we can adopt above surrogate objective function to improve sample efficiency.

### 3.3 ALGORITHM

In the following, we will present details about our proposed algorithm, which is referred as Context-aware Decentralized PPO (CD-PPO). We call it *context-aware* as the context $c_t^{-i}$ is a part of input to train policy and value functions of each agent. Such a context-aware approach can avoid the non-stationary problem occurred in independent learning methods (which does not consider dynamics of others during training), since the context reflects other agents' collective behaviors which can

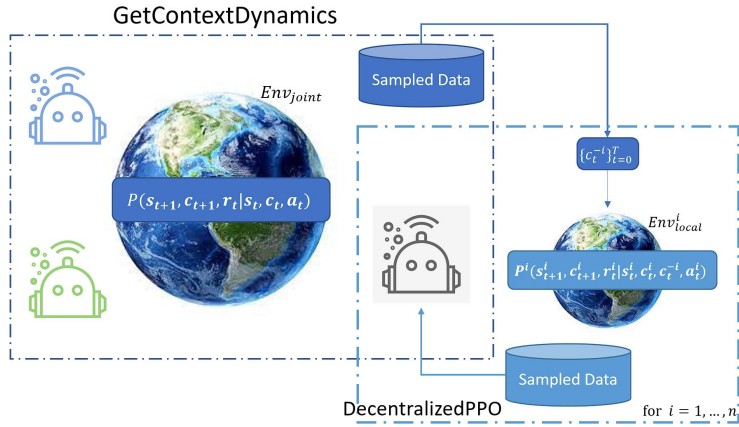

Figure 1: Our algorithm consists of two iterative learning procedures: 1. Get context dynamics $\{c_t^{-i}\}_{t=0}^T$ from the joint simulator and previous policies $\pi_{old}^i$, $\pi_{old}^{-i}$. 2. Train policy $\pi^i$ with data sampled from the local simulator conditioned on context dynamics $\{c_t^{-i}\}_{t=0}^T$.

impact dynamics of each individual agent. In the meanwhile, our method can mitigate the intractable centralized critic learning by avoiding using exploding joint state and action space, i.e., $(s_t, a_t, c_t)$, as input of the critic.

As shown in Algorithm 1, our algorithm consists of two iterative learning procedures: 1) obtaining context dynamics by running all agents in the joint environment (Algorithm 2); 2) updating policy of each agent using data sampled from its local simulator conditioned on the context dynamics (Algorithm 3). Moreover, our algorithm follows a decentralized training paradigm with an acceptable communication cost of the context dynamics. We refer readers to Appendix C for a detailed description.

It is worth noting that a naive approach is to optimize policies by Eq. 9 with data sampled from the joint simulator. Nonetheless, we find that it is empirically more time-consuming to sample one step in the joint simulator than letting each of the $n$ local simulators sample once. One major reason lies in that agent interactions take most of the simulation time. Our method, with the advantage of leveraging local simulators to simplify interactions, are henceforth much cheaper to sample. We refer readers to Appendix D for more discussions on the benefit of leveraging local simulators.

---

**Algorithm 1** Context-aware Decentralized PPO

---

Given the joint simulator $\text{Env}_{joint}$ and local simulators $\{\text{Env}_{local}^i\}_{i=1}^n$
Initialize policies $\pi^i$ and value functions $V^i$ for $i = 1, \ldots, n$
**for** $M$ epochs **do**
    // Collect context dynamics via running joint simulation
    $\{c_t^{-1}\}_{t=0}^T, \ldots, \{c_t^{-n}\}_{t=0}^T \leftarrow \textbf{GetContextDynamics}(\text{Env}_{joint}, \{\pi^i\}_{i=1}^n)$ (Algorithm 2)
    **for** $k = 1, 2 \ldots, K$ **do**
        **for all** agents $i$ **do**
            // Set capacity trajectory by context dynamics
            $\text{Env}_{local}^i.\text{set\_c\_trajectory}(\{c_t^{-i}\}_{t=0}^T)$
            // Train policy by running simulation in the corresponding local environment
            $\pi^i, V^i \leftarrow \textbf{DecentralizedPPO}(\text{Env}_{local}^i, \pi^i, V^i)$ (Algorithm 3)
        **end for**
    **end for**
    Evaluate policies $\{\pi^i\}_{i=1}^n$ on joint simulator $\text{Env}_{joint}$
**end for**

---

## 4 Experiments

We evaluate the performance of CD-PPO in three IM problems, which contain 5, 50, and 100 SKUs, respectively. By changing space size of the warehouse, we further testify how CD-PPO performs comparing to a series of baselines under different levels of competition. On these settings, we demonstrate that our algorithm can achieve comparable results as SOTA baselines, but is more sample-efficient.

### 4.1 Experiment setups

Our experiment is conducted on a simulator which can support both retailing and replenishment for multiple SKUs in one store. Instead of sampling demands from some hypothetical distributions, we instead directly use authentic demand data from Kaggle (Makridakis et al., 2020), which contains sales history of five years (2011-2016) for various SKUs in Walmart. We randomly choose 155 SKUs from the data and use sales of the first four years as our training set, while the others are the testing set. For all other information, e.g. price, cost, leading time etc., that are necessary to instantiate our simulator but not included in the data set, we will randomly sample them from certain reasonable ranges. The evaluation metric is the total profit in dollars. For all results that we present, we run each of all algorithms for four times with random seeds and present the average performance with standard deviations. The source code as well as instructions to reproduce our results can be found in `https://anonymous.4open.science/r/replenishment-marl-baselines-75F4`.

Note that the simulator we are using is developed for general purposes, it contains extra details like replenishing orders fulfillment scheduling, transportation management etc., hence it requires lots of computation resources to simulate IM problems with many SKUs ($> 100$). This also hinder us to extend our experiments to cases with more than 100 SKUs on a single machine. We leave it our future work to further testify our algorithm in a distributed environment where simulating large scale IM problems is possible.

We compare our method CD-PPO with the strong model-free on-policy MARL algorithms MAPPO (Yu et al., 2021) and IPPO (de Witt et al., 2020) on 5-SKUs environment and we do not show the performance of COMA (Foerster et al., 2018) and Value Decomposition methods such as QMIX (Rashid et al., 2018) due to their bad performance (i.e. negative profit). As for 50-SKUs and 100-SKUs scenario, we can only run IPPO, since MAPPO fails quickly as it suffers from the huge joint state-action space for training a centralized critic. Besides RL approaches, we also compare with base-stock policy, which is a well-known policy from OR community and widely adopted in practice (Kapuscinski & Tayur, 1999; Hubbs et al., 2020). (More details can be found in Appendix F). Our implementation is based on the EPyMARL (Papoudakis et al., 2021) framework, which contains MAPPO, IPPO and several common-used MARL algorithms.

As all agents are homogeneous in IM, we let parameters of policy network and critic network be shared amongst all agents. The two networks are all constructed by a two-layer MLP with hidden size 64. In our experiment, the actor network maps agent-specific state to a categorical distribution over a discrete action space $\{0, \frac{1}{3}, \frac{2}{3}, 1, \frac{4}{3}, \frac{5}{3}, 2, \frac{5}{2}, 3, 4, 5, 6, 7, 9, 12\}$, such that the real replenishment quantity is obtained by multiplying the action with an agent's sales mean of the past two weeks. In addition, to encourage policies to accommodate diverse context dynamics, we propose two methods to augment the context dynamics: 1) add noise to randomly chosen items with a predefined probability; 2) replace randomly chosen items with predicted values coming from a context generator model, which is also guided by the predefined probability. Besides, we also use the data collected in Algorithm 2 for policy training. More details about the algorithm can be found in Appendix C.

### 4.2 Main results

We evaluate CD-PPO, MAPPO and IPPO on 5,50,100-SKUs environments with different sizes of warehouse spaces. IPPO and MAPPO only use individual rewards rather than team reward (summation of all individual rewards) to train critics. For a fair comparison, we also train IPPO and MAPPO with information related to shared resource. The training curves for 5-SKUs scenario are

Table 1: Profit Comparison on Different Scenarios of 50 and 100 SKUs Environment

| Env_Scenario | CD-PPO(Ours) | IPPO-IR(w/o context) | IPPO-IR | IPPO(w/o context) | IPPO | Base-stock(Dynamic) |
|---|---|---|---|---|---|---|
| N50-C500 | $310.81 \pm 76.46$ | $235.09 \pm 60.61$ | $250.03 \pm 58.38$ | $164.43 \pm 143.01$ | $\mathbf{366.74 \pm 89.58}$ | $-408.14$ |
| N50-C2000 | $\mathbf{694.87 \pm 174.184}$ | $689.27 \pm 48.92$ | $545.86 \pm 459.71$ | $-1373.29 \pm 870.03$ | $-1102.97 \pm 1115.69$ | $42.71$ |
| N100-C1000 | $\mathbf{660.28 \pm 149.94}$ | $-2106.98 \pm 315.38$ | $-1126.42 \pm 409.83$ | $-1768.19 \pm 1063.61$ | $-669.83 \pm 1395.92$ | $-22.05$ |
| N100-C4000 | $\mathbf{1297.75 \pm 124.52}$ | $-2223.11 \pm 2536.00$ | $148.00 \pm 1017.47$ | $-6501.42 \pm 6234.06$ | $-6019.28 \pm 9056.49$ | $493.32$ |

shown in Figure 2. It is straightforward to observe that CD-PPO converges to the same performance comparing with other methods. In particular, CD-PPO is more sample-efficient due to its local simulations in parallel. To evaluate our algorithm on a larger scenario, we also run CD-PPO and IPPO (same with previous settings) on 50 and 100 SKUs and record the number of data samples when reaching the median performance of baselines. The results are summarized in Table 1 and Figure 3, where "N" and "C" denote the number of SKUs and the maximum capacity of shared resource, respectively. More details about the implementation and hyper-parameters we used can be found in Appendix C.

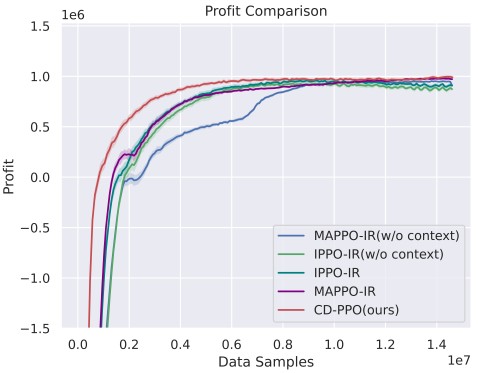

Figure 2: Training curves of different algorithms in 5-SKUs environment. "IR" means the environment only provides individual rewards and "w/o context" denotes the algorithm does not use information related to shared resource as parts of its inputs. The X-axis † also takes in samples from local simulation for CD-PPO.

Figure 3: Average samples needed by different algorithms in 100-SKUs environment to reach the median performance of baselines. The lower the value, the higher the sample-efficiency for the algorithm.

As demonstrated in Figure 2, Figure 3 and Table 1, CD-PPO is able to produce results comparable to other strong baselines and even continues improving as the training proceeds. Moreover, our algorithm can curtail non-stationary effects as in centralized training and, in the meanwhile, can scale up to scenarios with a large number of agents. In contrast, traditional MARL methods with CTDE paradigm even cannot start running on the 50-SKUs and the 100-SKUS environment, since the input of its critic is too huge to fit in the memory. IPPO, on the other hand, can run successfully, but lack stability under different levels of warehouse spaces. The full results and more ablation studies about how the capacity of shared resource affects the performance of CD-PPO and the influence of augmentation for context dynamics can be found in Appendix G.

## 5 RELATED WORK

In this section we will introduce relevant prior work including studies of IM problem and common-used training paradigms in MARL. More detailed description of other related work(e.g. Constrained/Weakly-Coupled MDP, Model-Based MARL and Mean-Filed MARL) are shown in Appendix A.

---

†Specifically, for one interaction in the global environment with N agents, we consider it as N data samples. For one interaction in the local simulator(based on context trajs) with one agent, we consider it as 1 data sample.

## 5.1 INVENTORY MANAGEMENT

Since the pioneer work in (Fukuda, 1964), many approaches have been proposed to solve different variants of IM problems, either using exact (Goldberg et al., 2016; Huh et al., 2009) or approximate (Halman et al., 2009; Fang et al., 2013; Chen & Yang, 2019) dynamic programming. As reinforcement learning based approaches are our main focus, we only present related work of this branch in the following. Interested readers will be referred to (Gijsbrechts et al., 2019) for an overview of traditional approaches for IM.

The attempt to apply reinforcement learning to solve inventory management problem has a long history, see for instance (Giannoccaro & Pontrandolfo, 2002; Jiang & Sheng, 2009; Kara & Dogan, 2018; Barat et al., 2019; Gijsbrechts et al., 2019; Oroojlooyjadid et al., 2017; 2020). However, as their main focus is to deal with challenges like volatile customer demands, bullwhip effects etc. in IM, they are restricted to simplified scenarios with only a single SKU. While these approaches are able to outperform traditional approaches in these scenarios, they overlook system constraints and coordination of SKUs imposed by shared resources. Exceptions are two recent works (Barat et al., 2019; Sultana et al., 2020), where more realistic scenarios containing multiple SKUs are considered. In (Barat et al., 2019), the main contribution is to propose a framework that supports efficient deployment of RL algorithms in real systems. As an example, the authors also introduce a centralized algorithm for solving IM problems. In contrast, a decentralized algorithm is proposed in (Sultana et al., 2020) to solve IM problems with not only multiple SKUs but also multiple echelons. In both works, the training algorithm is the advantage actor critic (A2C) (Wu et al., 2018).

## 5.2 TRAINING PARADIGM IN MARL

MARL algorithms generally fall between two frameworks: centralized and decentralized learning. There are two lines of fully decentralized training: Independent Learning methods (IL) (Tan, 1993; de Witt et al., 2020) and decentralized training with communication (Zhang et al., 2018; Sukhbaatar et al., 2016; Peng et al., 2017). For IL, each agent is learning independently to optimize its own reward and perceives the other agents as part of the environment. As for fully decentralized methods, representative methods usually build a direct communication pipe to share the message amongst agents to avoid non-stationary issue in MARL framework. One part of fully centralized approaches (Claus & Boutilier, 1998) assume a cooperative game and directly extend single-agent RL algorithms by learning a single policy to produce the joint actions of all agents simultaneously. Another type of centralized methods is called value decomposition(VD), which typically represents the joint Q-function as a function of agents' local Q-functions (Sunehag et al., 2017; Son et al., 2019; Rashid et al., 2018) and has been considered a gold standard in MARL. In contrast to previous methods, Centralised Training Decentralised Execution (CTDE) allows sharing of information during training, while policies are only conditioned on the agents' local observations enabling decentralised execution. The main category of CTDE algorithms are centralised policy gradient methods in which each agent consists of a decentralised actor and a centralised critic, which is optimised based on shared information between the agents. Representative studies of CTDE are MADDPG (Lowe et al., 2017), COMA (Foerster et al., 2018) and MAPPO (Yu et al., 2021) etc.

## 6 CONCLUSION

In this paper, we address inventory management problem for a single store with a large number of SKUs. Our method is based on shared resource structure, where agents only interact indirectly with each other through shared-resource. By leveraging such structure, our method can outperform existing MARL algorithms in terms of both final performance and computation efficiency. It is worth mentioning that our method is not limited to IM, but also applicable to a wide range of real-world applications with shared-resource structure.

In real-world applications, we usually need to deal with thousands of agents, which poses a challenge for existing MARL algorithms. To address such challenges, we need develop efficient algorithms which are compatible with distributed training. In this paper, we take our first step towards developing efficient and practical MARL algorithms for real-world applications with shared-resource structure, and will continue to address above challenges arisen in real-world applications in our future work.

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

# A RELATED WORK

## A.1 TRAINING PARADIGM IN MARL

In this section we will provide a more detailed description of related work for training paradigm used in MARL. From the perspective of training schemes, it can be devided into three categories: decentralized training decentralized execution (DTDE), centralized training centralized execution (CTCE) and centralized training decentralized execution (CTDE). Recent deep MARL works often use the CTDE or CTCE training pradigm.

The CTCE paradigm allows the straightforward employment of single-agent approaches such as Actor-Critic (Mnih et al., 2016) or policy gradient algorithms (Schulman et al., 2017) to multi-agent problems. The representative work of CTCE is (Gupta et al., 2017), which represented the centralized executor as a set of independent sub-policies such that agents' individual action distributions are captured rather than the joint action distribution of all agents.
Value-based CTDE approaches, which are also known as Value Decomposition methods (Peng et al., 2020; Mahajan et al., 2019; Rashid et al., 2020; 2018; Son et al., 2019; Sunehag et al., 2017; Wang et al., 2020b; 2019), mianly focus on how centrally learned value functions can be reasonably decoupled into decentralized ones and have shown promising results. Policy-gradient-based methods on CTDE, on the other hand, have heavily relied on centralized critics. One of the first works utilizing a centralized critic was COMA (Foerster et al., 2018), a framework adopting a centralized critic with a counterfactual baseline. For convergence properties, COMA establishes that the overall effect on decentralized policy gradient with a centralized critic can be reduced to a single-agent actor-critic approach, which ensures convergence under the similar assumptions like A2C.

Concurrently with COMA, MADDPG (Lowe et al., 2017) proposed to use a dedicated centralized critic for each agent in semi-competitive domains, demonstrating compelling empirical results in continuous action environments. Recently, MAPPO (Yu et al., 2021), an on-policy policy gradient multi-agent reinforcement learning algorithm, achieves strong results comparable to the state-of-the-art on a variety of cooperative multi-agent challenges. Despite its on-policy nature, MAPPO is competitive to ubiquitous off-policy methods such as MADDPG, QMix, and RODE in terms of final performance, and in the vast majority of cases, comparable to off-policy methods in terms of sample-efficiency. In addition, many incremental research inspired by MADDPG, COMA or MAPPO also borrowed the centralized critic baselines e.g. M3DDPG (Li et al., 2019), SQD-DPG (Wang et al., 2020a), etc. Mean Filed Q-Learning (Yang et al., 2018; Carmona et al., 2019) takes a different approach from the CTDE based methods. It employs mean field approximation over the joint action space in order to address the scalability issue that exists in the prior methods.

Contrary to CTDE, in DTDE paradigm, each agent has an associated policy which maps local observations to a distribution over individual actions. No information is shared between agents such that each agent learns independently. DTDE has been applied to cooperative navigation task (Chen et al., 2017; Strouse et al., 2018), to partially observable domains (Dobbe et al., 2017; Srinivasan et al., 2018), and to social dilemmas (Leibo et al., 2017). For more comparisons of centralized critic and decentralized critic, please see (Lyu et al., 2021).

In this paper, we design a decentralized traning paradigm avoiding the flaws of traditional training paradigms proposed in literature. The fundamental drawback of the DTDE paradigm is that the environment appears non-stationary from a single agent's viewpoint because agents neither have access to the knowledge of others, nor do they perceive the joint action. Some studies reported that DTDE scale poorly with the number of agent due to the extra sample complexity, which is added to the learning problem (Gupta et al., 2017); An obvious flaw for CTDE/CTCE is that state-action spaces grow exponentially by the number of agents. Even though there are some attempts proposed that the joint model can be factored into individual policies for each agent To address the so-called curse of dimensionality, CTDE/CTCE methods have to use at least joint states overall agents as the input to approximate global value function to give guidance for centralized critics or decentralized policies. Meaning that these traditional training schemes still not have strong expansion capabilities

to large number of agents when system's state is combined by local state for each agent. As for our approach, we train agents independently with learned dynamics model of utilization's trend for shared resource, which will give agent enough information to learn optimal policy (we will explain it in the following sections). At the meantime, we can also, improve efficiency of data sampling since we don't always use the original joint simulator containing all agents for data collection. Instead, we mainly running the light-cost local simulator by embedding the learned dynamic model, which can significantly reduce the cost of data collection process, especially when running joint simulator (such as inventory management) expensively.

### A.2 MDP SCENARIOS IN MARL

There are limited similar variants of MDP settings which have been studied under MARL framework, to our knowledge. Dec-POMDP (Oliehoek & Amato, 2016) is the most common setting studied in MARL research, especially in fully cooperative tasks in which agents share the global team reward rather than individual rewards; In the scenarios of Constrained MDP setting (C-MDP, (Bhatnagar & Lakshmanan, 2012; Wang et al., 2019; Diddigi et al., 2019)), there are some constraints in the system, such as the penalty caused by illegal actions in autonomous driving (Zhang et al., 2016; Fowler et al., 2018), the resource-allocation constraints in scheduling tasks (Agrawal et al., 2016; Dolgov & Durfee, 2006), or limited bandwidth during communicating among agents (Fowler et al., 2018), etc. Similar with C-MDP, Weakly-Coupled Constrained MDP (WC-C-MDP (Boutilier & Lu, 2016)) consider the problem of budget (or other resource) allocation in sequential decision problems involving a large number of concurrently running sub-processes,whose only interaction is through their consumption of budget. Different from mentioned scenarios, we focus on the situation where agents can only get the individual reward from the environment, so that team reward is invisible; Comparing with C-MDP and WC-C-MDP, the penalty of overstocking in which is related with stock levels of all agents in future, while in C-MDP and WC-C-MDP, the corresponding cost function is only based on historical states of the system. Leading that we can not optimize only the combined objective of long-term summation of return and penalty.

### A.3 MODEL-BASED MARL

For model-based MARL, there are relatively limited works of literature. P-MARL (Marinescu et al., 2015) proposed an approach that integrates prediction and pattern change detection abilities into MARL and thus minimises the effect of non-stationarity in the environment. The environment is modelled as a time-series, with future estimates provided using prediction techniques. Learning is based on the predicted environment behaviour, with agents employing this knowledge to improve their performance in real-time. (Park et al., 2019) proposed to use a centralized auxiliary prediction network to model the environment dynamics to alleviate the non-stationary dynamics problem. (Krupnik et al., 2020) built a centralized multi-step generative model with a disentangled variational auto-encoder to predict the environment dynamics and the opponent actions and then performed trajectory planning. AORPO (Zhang et al., 2021) is a Dyna-style method in which each agent builds its multi-agent environment model that consist of a dynamics model and multiple opponent models, and trains its policy with the data both generated from adaptive opponent-wise rollout and interacted from the real environment. To our knowledge, unlike previous work, our approach is the first to accelerate the learning process in MARL by building only a part dynamic model of the whole system. Our algorithm is also a Dyna-style method like AORPO. And it is clear that the difficulty of modeling process will be reduced significantly and the learning process is more efficient.

### A.4 MEAN-FIELD MARL

## B DETAILS FOR INVENTORY MANAGEMENT PROBLEM

We summarize all notations for Section 2.2 in Table 2. We also give an example with 2 SKUs in Fig. 4 to further illustrate the whole procedure for inventory dynamics.

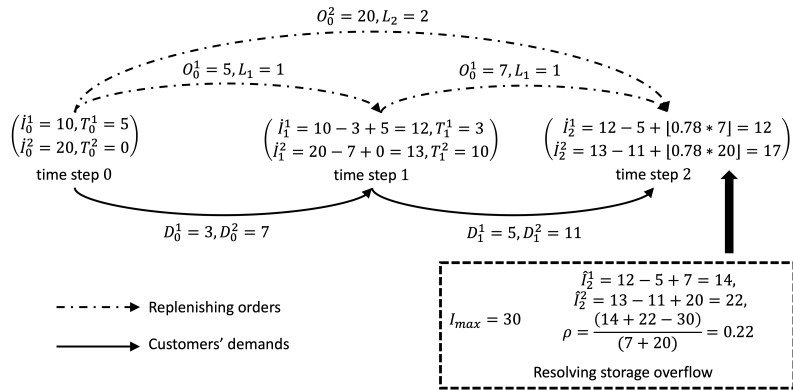

Figure 4: A diagram to illustrate an inventory dynamics in two time steps.

Table 2: Notations.

| Notation | Explanation |
|---|---|
| $\check{I}_t^i$ | Units in stock of SKU $i$ at the $t$-th time step |
| $\hat{I}_t^i$ | Units in stock of SKU $i$ at the end of time step $t$ if no discard |
| $O_t^i$ | Order quantity of the $i$-th SKU at the $t$-th time step |
| $D_t^i$ | Demand of the $i$-th SKU at the $t$-th time step |
| $S_t^i$ | Sale quantity of the $i$-th SKU at the $t$-th time step |
| $T_t^i$ | Units in transit of the $i$-th SKU at the $t$-th time step |
| $Pt_t^i$ | Profit generated on the $i$-th SKU at the $t$-th time step |
| $p_i$ | Unit sales price of $i$-th SKU |
| $q_i$ | Unit procurement cost of $i$-th SKU |
| $o$ | Unit order cost |
| $h$ | Unit holding cost for a time step |

## C  ALGORITHM DETAILS

Here we provide the pseudocode of our algorithm with context augmentation in Algorithm 4. And details of sub-algorithms will be introduced in the following subsections. Notes that all pseudocode are assumed using RNN as networks.

### C.1  GET CONTEXT DYNAMICS

Our algorithm follows the algorithmic structure of the Multi-Agent PPO (Yu et al., 2021), in which we maintain two separate networks for $\pi_\theta$ and $V_\phi(s)$. And the first stage of our algorithm is only running policies in the origin joint environment to get episodes for context dynamics, i.e. the trajectories of $c_t^i$ and $c_t^{-i}$. This process is similar with all other traditional MARL methods. Notes that we can also save the transitions of each agent for learning policy and value function network. In the following pseudocode of joint sampling(Algorithm 2), we only record the part for the context dynamics to train the context model prepared for next stage's training.

After data collecting for context dynamics, we will use a LSTM model $f_\mathbf{c}$ to train a surrogate predictor as an extra augmentation for collected dynamics in next stage. In details, we split the collected trajectories of context into several sub-sequences of length 8 and the training objective is to minimize the mean-squared error of the $L + 1$ day's capacity predicted given the dynamics of previous $L$ days.

$$\min(f_\mathbf{c}(c_{t-L}, \ldots, c_t; \omega) - c_{t+1})^2 \tag{14}$$

## C.2 DECENTRALIZED PPO

With the collected context dynamics of the shared resource, it is easy to run the second stage: sampling on the local simulators for each agent and then training the policy and critic with data. It is worth noting that the main difference between our training paradigm and traditional MARL methods under CTDE structure is that we directly sampling local observations in the extra simulators in which only one agent existed rather than the joint simulator in which all agents interacting with each other. In other words, in the new local simulator, there is only one SKU in the entire store, and the trend of available capacity is completely simulated according to the given context dynamics.

In practice, we parallelly initialize new instances of the original inventory environment with the new configure settings which only contains a specific SKU $i$ and a fixed trajectory of context. As for the fixed context trajectory, we use the subtraction results $\{c^{-i}\}_{t=0}^T$ with some augmentations: 1) add some noise in some items with a predefined probability;2) replace some items with predicted values comes from the trained context model also by the predefined probability. Then we run the policy to interact with the local simulators to conduct episodes under the embedded context dynamics and put them into a shared replay buffer since all transitions are homogeneous and we shared the parameters over all policies. And decentralized training will be started by utilizing the shared replay buffer of transitions collected from local simulators.

We consider a variant of the advantage function based on decentralized learning, where each agent learns a agent-specific state based critic $V_\phi(s_t^i)$ parameterised by $\phi$ using *Generalized Advantage Estimation* (GAE, (Schulman et al., 2016)) with discount factor $\gamma = 0.99$ and $\lambda = 0.95$. We also add an entropy regularization term to the final policy loss (Mnih et al., 2016). For each agent $i$, we have its advantage estimation as follows:

$$A_t^i = \sum_{l=0}^h (\gamma\lambda)^l \delta_{t+l}^i \tag{15}$$

where $\delta_{ti}^i = r_t\left(s_t^i, a_t^i\right) + \gamma V_\phi\left(z_{t+1}^i\right) - V_\phi\left(s_t^i\right)$ is the TD error at time step $t$ and $h$ is marked as steps num. And we use individual reward provided from local simulator $r_t^i(s_t^i, a_t^i)$. So that the final policy loss for each agent $i$ becomes:

$$\mathcal{L}^i(\theta) = \mathbb{E}_{s_t^i, a_t^i \sim \mathcal{T}_{\text{local}}(c_t^{-i})} \left[ \min\left( \frac{\pi_\theta\left(a_t^i \mid s_t^i\right)}{\pi_{\theta_{old}}\left(a_t^i \mid s_t^i\right)} A_t^i, \right.\right.$$
$$\left.\left. \text{clip}\left( \frac{\pi_\theta\left(a_t^i \mid s_t^i\right)}{\pi_{\theta_{old}}\left(a_t^i \mid s_t^i\right)}, 1 - \epsilon, 1 + \epsilon \right) A_t^i \right)\right] \tag{16}$$

As for training value function, in addition to clipping the policy updates, our method also use value clipping to restrict the update of critic function for each agent $i$ to be smaller than $\epsilon$ as proposed by GAE using:

$$\mathcal{L}^i(\phi) = \mathbb{E}_{s_t^i \sim \mathcal{T}_{\text{local}}(c_t^{-i})} \left[ \min\left\{ \left(V_\phi\left(s_t^i\right) - \hat{V}_t^i\right)^2, \right.\right.$$
$$\left.\left. \left(V_{\phi_{old}}\left(s_t^i\right) + \text{clip}\left(V_\phi\left(s_t^i\right) - V_{\phi_{old}}\left(s_t^i\right), -\epsilon, +\epsilon\right) - \hat{V}_t^i\right)^2 \right\}\right] \tag{17}$$

where $\phi_{old}$ are old parameters before the update and $\hat{V}_t^i = A_t^i + V_\phi\left(s_t^i\right)$. The update equation restricts the update of the value function to within the trust region, and therefore helps us to avoid overfitting to the most recent batch of data. For each agent, the overall learning loss becomes:

$$\mathcal{L}(\theta, \phi) = \sum_{i=1}^N \mathcal{L}^i(\theta) + \lambda_{\text{critic}}\mathcal{L}^i(\phi) + \lambda_{\text{entropy}}\mathcal{H}\left(\pi^i\right) \tag{18}$$

It is obvious that all networks are trained in the decentralized way since their inputs are all local variables which stem from the light-cost local simulators. As mentioned before, at this learning

stage, there are no interactions between any two agents. Although it seems like the way of independent learning, we need to point that we use the global context simulated from the joint environment, which is essentially different from independent learning methods since they will not consider this style global information which is simulated from joint simulator but be fixed in the local simulators. Our decentralized training have several advantages: firstly, the local simulator is running efficient because of its simple only-one-agent transition function; secondly, this paradigm avoid the issue for non-stationary occurred in the traditional MARL methods since there are no interaction amongst agents so that it is no need to consider influences of other agents; thirdly, we can use more diverse context trajectories to make agents face various situations of the available levels of the store, which leads to improve the generalization of the networks to be trained; fourthly, it is easy for this training paradigm to be extended to large-scale distributed training by running parallel simulation whose communication cost is also acceptable for modern distributed training frameworks.

If the critic and actor networks are RNNs, then the loss functions additionally sum over time, and the networks are trained via Backpropagation Through Time (BPTT). Pseudocode for local sampling with recurrent version of policy networks is shown in Algorithm 3.

---

**Algorithm 2** GetContextDynamic

---

   **INPUT** policies $\{\pi^i_{\theta_i}\}^n_{i=1}$ and the joint simulator $\text{Env}_{joint}$
   (Optional) Initialize $\omega$, the parameters for context model $f_{\mathbf{c}}$
   set data buffer $D = \{\}$
   **for** $bs = 1$ **to** $batch\_size$ **do**
      $\tau = []$ empty list
      initialize $h^1_{0,\pi}, \cdots h^n_{0,\pi}$ actor RNN states
      **for** $t = 1$ **to** $T$ **do**
         **for all** agents $i$ **do**
            $p^i_t, h^i_{t,\pi} = \pi^i(s^i_t, c_t, h^i_{t-1,\pi}; \theta_i)$
            $a^i_t \sim p^i_t$
         **end for**
         Execute actions $\boldsymbol{a_t}$, observe $\boldsymbol{r_t}, \boldsymbol{s_{t+1}}$
         $\tau += [\boldsymbol{s_t}, c_t, \boldsymbol{h_{t,\pi}}, \boldsymbol{a_t}, \boldsymbol{r_t}, \boldsymbol{s_{t+1}}, c_{t+1}]$
      **end for**
      // Split trajectory $\tau$ into chunks of length L
      **for** l = 0, 1, .., T//L **do**
         $D = D \cup (c[l : l + T])$
      **end for**
   **end for**
   // (Optional) Train the context model for augmentation
   **if** Need to train context model **then**
      **for** mini-batch $k = 1, \ldots, K_1$ **do**
         $b \leftarrow$ random mini-batch from D with all agent data
         Update capacity-context dynamics model with Eq. 14
      **end for**
      Adam update $\omega$ with data $b$
   **end if**
   **OUTPUT** context dynamics $\{c^{-i}_t\}^T_{t=1}$ for $i = 1, \ldots, n$; (Optional) $f_\omega$

---

## D   LOCAL SAMPLING V.S. JOINT SAMPLING

Our algorithm follows a decentralized training paradigm, which is compatible with distributed training framework. We can run parallel local simulation while communication cost of context dynamics is also acceptable. With distributed training, our method can be much more efficient than learning from the joint simulator directly.

One may argue that we can also learn from joint simulator efficiently by implementing a distributed/parallelized joint simulator. While this can indeed improve sampling efficiency in many scenarios, its usefulness is limited particularly for systems with a large number of agents. In a typ-

---

**Algorithm 3** DecentralizedPPO

---

// Generate data for agent $i$ with corresponding context dynamic model
**INPUT** local simulator $\text{Env}_{local}^i$, policy $\pi_{\theta_i}^i$ and value function $V_{\phi_i}^i$
Set data buffer $D = \{\}$
Initialize $h_{0,\pi}^1, \cdots, h_{0,\pi}^n$ actor RNN states
Initialize $h_{0,V}^1, \ldots h_{0,V}^n$ critic RNN states
$\tau = []$ empty list
**for** $t = 1$ **to** $T$ **do**
    $p_t^i, h_{t,\pi}^i = \pi^i(s_t^i, h_{t-1,\pi}^i, c_t^i; \theta_i)$
    $a_t^i \sim p_t^i$
    $v_t^i, h_{t,V}^i = V^i(s_t^i, c_t^i, h_{t-1,V}^i; \phi_i)$
    Execute actions $a_t^i$ in $\text{Env}_{local}^i$, and then observe $r_t^i, c_{t+1}^i, s_{t+1}^i$
    $\tau += [s_t^i, c_t^i, a_t^i, h_{t,\pi}^i, h_{t,V}^i, s_{t+1}^i, c_{t+1}^i]$
**end for**
Compute advantage estimate $\hat{A}$ via GAE on $\tau$ (Eq. 15)
Compute reward-to-go $\hat{R}$ on $\tau$
// Split trajectory $\tau$ into chunks of length L in $D$
**for** l = 0, 1, .., T//L **do**
    $D = D \cup (\tau[l : l + T, \hat{A}[l : l + L], \hat{R}[l : l + L])$
**end for**
**for** mini-batch $k = 1, \ldots, K_2$ **do**
    $b \leftarrow$ random mini-batch from D with all agent data
    **for** each data chunk $c$ in the mini-batch $b$ **do**
        update RNN hidden states for $\pi^i$ and $V^i$ from first hidden state in data chunk
    **end for**
**end for**
Calculate the overall loss according to Eq. 16 to Eq. 18
Adam update $\theta_i$ on $L^i(\theta_i)$ and $\mathcal{H}$ with data $b$
Adam update $\phi_i$ on $L^i(\phi_i)$ with data $b$
**OUTPUT** policy $\pi_{\theta_i}^i$ and value function $V_{\phi_i}^i$

---

**Algorithm 4** Context-aware Decentralized PPO with Context Augmentation

---

Given the joint simulator $\text{Env}_{joint}$ and local simulators $\{\text{Env}_{local}^i\}_{i=1}^n$
Initialize policies $\pi^i$ and value functions $V^i$ for $i = 1, \ldots, n$
Initialize context model $f_\omega$ and the augmentation probability $p_{aug}$
**for** $M$ epochs **do**
    // Collect context dynamics via running joint simulation
    $\{c_t^{-1}\}_{t=0}^T, \ldots, \{c_t^{-n}\}_{t=0}^T, f_\omega \leftarrow$ **GetContextDynamics**$(\text{Env}_{joint}, \{\pi^i\}_{i=1}^n, f_\omega)$ (Algorithm 2)

    **for** $k = 1, 2 \ldots, K$ **do**
        **for all** agents $i$ **do**
            // Set capacity trajectory by augmented context dynamics
            $\text{Env}_{local}^i.\text{set\_c\_trajectory}(aug(\{c_t^{-i}\}_{t=0}^T, f_\omega, p_{aug}))$
            // Train policy by running simulation in the corresponding local environment
            $\pi^i, V^i \leftarrow$ **DecentralizedPPO**$(\text{Env}_{local}^i, \pi^i, V^i)$ (Algorithm 3)
        **end for**
    **end for**
    Evaluate policies $\{\pi^i\}_{i=1}^n$ on joint simulator $\text{Env}_{joint}$
**end for**

---

ical multi-agent system, interactions occur frequently among all agents. Actually, one advantage of multi-agent systems is to model complex interactions among agents. However, such interactions will remarkably reduce the efficiency improvement brought by parallelism, as these interactions often require involved agents to synchronize which usually consumes lots of time on waiting and communicating. For instance, in IM problems, at each time step, all agents should be synchronized in order to calculate $\rho$ in Eq. 6 before moving to the next time step.

On the other hand, in our local sampling approach we simplify such costly interactions by utilizing special structures of shared-resource stochastic games. Under our approach, there is no need for agents to be synchronized before moving to the next time step in local simulators. As a result, our method can be much more efficient than learning only from the joint simulator in practice.

# E TRAINING DETAILS

## E.1 THE CODEBASE

As part of this work we extended the well-known EPyMARL codebase((Papoudakis et al., 2021)) to integrated our simulator and algorithm, which already include several common-used algorithms and support more environments as well as allow for more flexible tuning of the implementation details. It is convenience for us to compare our algorithm with other baselines. All code for our new codebase is publicly available open-source on Anonymous GitHub under the following link: `https://anonymous.4open.science/r/replenishment-marl-baselines-75F4`

## E.2 HYPERPARAMETERS DETAILS

Table 3 presents the hyperparameters used in our algorithm for 5-SKUs environment.

| Hyperparameters | Value |
|---|---|
| runner | ParallelRunner |
| batch size run | 10 |
| decoupled training | True |
| use individual envs | True |
| max individual envs | 5 |
| decoupled iterations | 1 |
| train with joint data | True |
| context perturbation prob | 1.0 |
| hidden dimension | 64 |
| learning rate | 0.00025 |
| reward standardisation | True |
| network type | FC |
| entropy coefficient | 0.01 |
| target update | 200 (hard) |
| n-step | 5 |

Table 3: Hyparameters used in CD-PPO

## E.3 DETAILS OF STATES AND REWARDS

Table 4 shows the features of the state for our MARL agent that corresponds to the $i$-th SKU on the $t$-th time step.

It's worthy to note that we use the profit generated on the $i$-th SKU at the $t$-th time step $Pt_t^i$ divided by 1000000 as the individual reward of $i$-th agent at the $t$-th time step, for team reward methods, we simply sum up all the individual rewards, which corresponds to the daily profit of the whole store at the $t$-th time step, divided by 1000000.

| | Features |
|---|---|
| **State** | |
| Storage information | Storage capacity C |
| Inventory information | Quantity of products in stock $I_t^i$ |
| | Quantity of products in transit $T_t^i$ |
| History information | Replenishment history in the latest 21 days $\left[O_{t-21}^i, \cdots, O_{t-1}^i\right]$ |
| | Sales history in the latest 21 days $\left[S_{t-21}^i, \cdots, S_{t-1}^i\right]$ |
| | Standard deviation of historical sales $\mathrm{std}\left(S_{t-21}^i, \cdots, S_{t-1}^i\right)$ |
| Product information | Unit sales price $p_i$ |
| | Unit procurement cost $q_i$ |
| **Context** | |
| Global storage utilization | Current total storage level of the store $\sum_{j=1}^{l-1} I_t^j$ |
| Global unloading level | Current total unloading level of the store $\sum_{j=1}^{l-1} O_{t-L_j+1}^j$ |
| Global excess level | Current total excess level of the store $\rho \times \sum_{j=1}^{l-1} O_{t-L_j+1}^j$ |

Table 4: Features of the state

## F BASE-STOCK POLICY

---

**Algorithm 5** Base-stock Policy

---

**Input:**
$\{D_t^i\}_{t=t_1}^{t_2}$, $v$, $\tau$, IM parameters
**Output:**
$Z_i, \{O_t^i\}_{t=t_3}^{t_4}$
// Description:Base-stock policy for single SKU i
// $Z_i$:Base-stock level for SKU i
// $\{O_t^i\}_{t=t_3}^{t_4}$:Base-stock replenishment policy for SKU i, from $t_3$ to $t_4$
// $\{D_t^i\}_{t=t_1}^{t_2}$:Demand series of SKU i used to infer $Z_i$, from $t_1$ to $t_2$
// $v : v \geq 1, v \in \mathbb{R}$, Hyper-parameter to control storage utilization level
// $\tau : \tau \in \mathbb{N}_+$, Hyper-parameter to control replenishing interval
// IM parameters:including leading time $L_i$, storage capacity $C$, etc

// Solve Mixed Integer Programming with dual simplex methods:

$$Z_i \leftarrow \max_{Z_i} \sum_{t=t_1}^{t_2} Pt_t^i \text{ s.t.} \begin{cases} S_t^i = \min\left(D_t^i, I_t^i\right) \\ T_{t+1}^i = T_t^i - O_{t-L_i+1}^i + O_{t+1}^i \\ I_{t+1}^i = I_t^i - S_t^i + O_{t-L_i+1}^i \\ O_t^i = \max\left(0, Z_i - I_t^i - T_t^i\right) \\ Pt_t^i = p_i S_t^i - q_i O_t^i - h I_t^i \\ t_1 \leq t \leq t_2 \end{cases}$$

// Replenishing policy deduction:
$O_t^i = \max\left(0, Z_i - I_t^i - T_t^i\right), t_3 \leq t \leq t_4$
$O_t^i = \min\left(O_t^i, vC - \sum_{j=1}^n (I_t^j + T_t^j)\right), t_3 \leq t \leq t_4$
$O_t^i = O_t^i \mathbb{I}\left[t \bmod \tau = 0\right], t_3 \leq t \leq t_4$
return $Z_i, \{O_t^i\}_{t=t_3}^{t_4}$

---

In addition to comparing with related MARL baselines, we also add a well-known algorithm "Base-stock" from OR community as our non-RL baseline. The pseudo-code for base-stock policy can be found in Algorithm 5, where $Z_i$ is called the base-stock for agent $i$ and is computed by solving a mixed integer programming problem. After that, $Z_i$ will be used to guide replenishment of agent $i$ periodically. We shall note that base-stock policy can not deal with complex IM variant settings

like coordinated replenishment (multiple SKUs with storage capacities), order cost and stochastic VLTs, etc. These complicated realistic constraints are exactly what we use to test other MARL algorithms. Thus it may happen that Base-stock policies constantly overflow warehouse capacities when storage is tight, in which cases incoming products are abandoned proportionally as we explained in Section 2.2. This explains Base-stock's poor performance on some of the envs.

Base-stock utilizes Linear Programming to work out the base stock levels and thus the replenishing policy. "Static" indicates that the base stock levels are based on demand data from training set and then kept static while making decisions on test set. "Dynamic" updates its base stock levels on a regular time cycle. "Oracle" directly access the whole test set to calculate its base stock levels, which is essentially a cheating version used to show the upper limits of Base-stock policy. In practice, we conduct grid search to find proper $v$ and $\tau$. We refer readers to (Hubbs et al., 2020) for a detailed description. Our implementation of Base-stock baselines is also inspired by it.

## G  ADDITIONAL RESULTS

### G.1  THE FULL RESULTS ON ALL ENVIRONMENTS

Here we present all the experiment results in Table 5 and display the training curves of algorithms in 50-SKUs scenarios in Figure 5 and Figure 6. We also present the results from OR methods designed to deal with IM problem, namely Base-stock, in Table 5.

Please note that, actually, VLT (i.e leading time) for each SKU in N50 and N100 is stochastic during simulation. On the one hand, the specified VLTs can be modelled as exponential distributions in our simulator. On the other hand, the real lead time for each procurement can be longer than specified due to lack of upstream vehicles. (Distribution capacity is also considered in the simulator, though not the focus of this paper). So that the results running on N50 and N100 are all considering stochastic VLTs.

Table 5: Profit Comparison on All Environments

| | **Profit(10k dollar)** | | | | | | | | | | | |
|---|---|---|---|---|---|---|---|---|---|---|---|---|
| Env Scenario | CD-PPO(ours) | IPPO-IR(w/o context) | MAPPO-IR(w/o context) | IPPO-IR | MAPPO-IR | IPPO(w/o context) | MAPPO(w/o context) | IPPO | MAPPO | Basestock(Static) | Basestock(Dynamic) | Basestock(Oracle) |
| N5-C50 | 40.58 ± 6.02 | 40.37 ± 4.89 | 39.32 ± 15.53 | 43.33 ± 3.30 | 54.87 ± 9.26 | **74.11 ± 1.55** | 49.24 ± 1.32 | 63.22 ± 13.75 | 48.49 ± 1.89 | 17.4834 | 33.9469 | 38.6207 |
| N5-C100 | **99.21 ± 1.91** | 92.41 ± 2.78 | 94.70 ± 18.84 | 91.38 ± 3.57 | 97.69 ± 14.41 | 97.89 ± 6.65 | 74.71 ± 1.51 | 92.90 ± 13.36 | 71.57 ± 3.14 | 48.8944 | 80.8602 | 97.7010 |
| N50-C500 | 310.81 ± 76.46 | 235.09 ± 60.61 | N/A | 250.03 ± 58.38 | N/A | 164.43 ± 143.01 | N/A | **366.74 ± 89.58** | N/A | −430.0810 | −408.1434 | −397.831 |
| N50-C2000 | **694.87 ± 174.184** | 689.27 ± 48.92 | N/A | 545.86 ± 459.71 | N/A | −1373.29 ± 870.03 | N/A | −1102.97 ± 1115.69 | N/A | −15.5912 | 42.7092 | 1023.6574 |
| N100-C1000 | **660.28 ± 149.94** | −2106.98 ± 315.38 | N/A | −1126.42 ± 409.83 | N/A | −1768.19 ± 1063.61 | N/A | −669.83 ± 1395.92 | N/A | −173.39 | −22.05 | 91.17 |
| N100-C4000 | **1297.75 ± 124.52** | −2223.11 ± 2536.00 | N/A | 148.00 ± 1017.47 | N/A | −6501.42 ± 6234.06 | N/A | −6019.28 ± 9056.49 | N/A | 410.59 | 493.32 | 755.47 |

Table 6: Average samples needed by different algorithms to reach the median performance of baselines on All Environments

| | **Data Samples(10k)** | | | | | | | | |
|---|---|---|---|---|---|---|---|---|---|
| Env Scenario | CD-PPO(ours) | IPPO-IR(w/o context) | MAPPO-IR(w/o context) | IPPO-IR | MAPPO-IR | IPPO(w/o context) | MAPPO(w/o context) | IPPO | MAPPO |
| N5-C50 | ∞ | ∞ | ∞ | ∞ | 708.10 | **588.63** | 1671.10 | 708.10 | 1671.10 |
| N5-C100 | **522.80** | ∞ | 711.97 | ∞ | 987.27 | 806.56 | ∞ | 1298.40 | ∞ |
| N50-C500 | **5484.49** | ∞ | N/A | ∞ | N/A | 9195.23 | N/A | 12802.89 | N/A |
| N50-C2000 | **2996.87** | 3138.49 | N/A | 8483.04 | N/A | ∞ | N/A | ∞ | N/A |
| N100-C1000 | **47.14** | ∞ | N/A | ∞ | N/A | 539.26 | N/A | 191.88 | N/A |
| N100-C4000 | **60.07** | ∞ | N/A | 127.57 | N/A | ∞ | N/A | 1151.57 | N/A |

It's worthy noting that throughout all the environments we've experimented on, CD-PPO enjoys higher sample efficiency as shown in Table 6. Though it seems that in environment where the storage capacity is extremely tense, like N5-C50, CD-PPO performs not as well as IPPO, we reason that these tense situations favor IPPO (with team reward), which may quickly learn how to adjust each agents' behavior to improve team reward. While CD-PPO, owing to its decentralized training paradigm, struggles to learn a highly cooperative policy to cope with the strong resource limits while meeting stochastic customer demands. Yet we still claim that CD-PPO does outperform its IR (w/o context) baselines while producing comparable performance when compared to IR (with context) algorithms. On the other hand, IPPO(with team reward) struggles to learn a good policy in 50-agents environment for the large action space compared to the single team reward signal, and it fails when faced with N50-C2000 whose state space is even larger. To top it all off, CD-PPO does have its strengths in terms of sample efficiency. Though we will continue to address the above challenge in our future work.

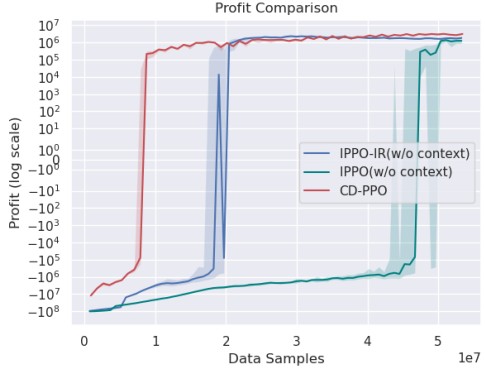 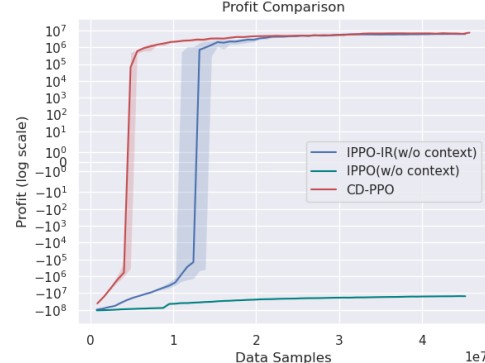

Figure 5: Training curves on N50-C500.  Figure 6: Training curves on N50-C2000

## G.2 ABLATION STUDIES FOR CONTEXT AUGMENTATION

In this section, we aim to seek a better way to augment the context dynamics in order to boost performance of our algorithm. We ponder the following two questions:

**Q1: Which is a better way to augment the original data, adding noise or using a Deep prediction model?**

To answer this question, we set out experiments on environment N5-C100 with our algorithm CD-PPO, using either context trajectories generated by a deep LSTM prediction model, or simply adding a random normal perturbation to the original dynamics trajectories, both on 3 different seeds. As shown in Figure 7, those runs with deep prediction model generated dynamics enjoy less std and better final performance. This could result from that the diversity of deep model generated trajectories surpasses that of random perturbation.

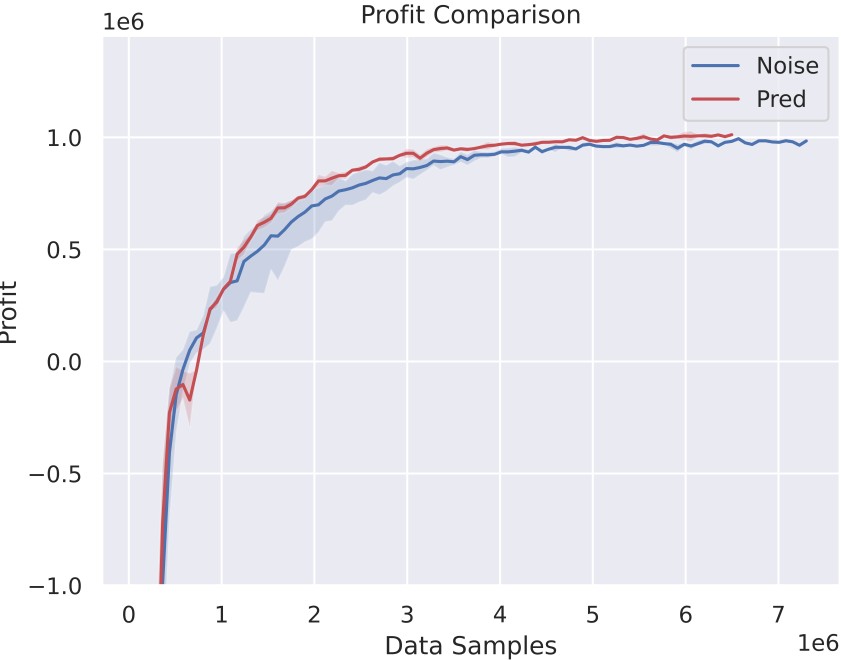

Figure 7: Training curves of CD-PPO with different augmentation methods

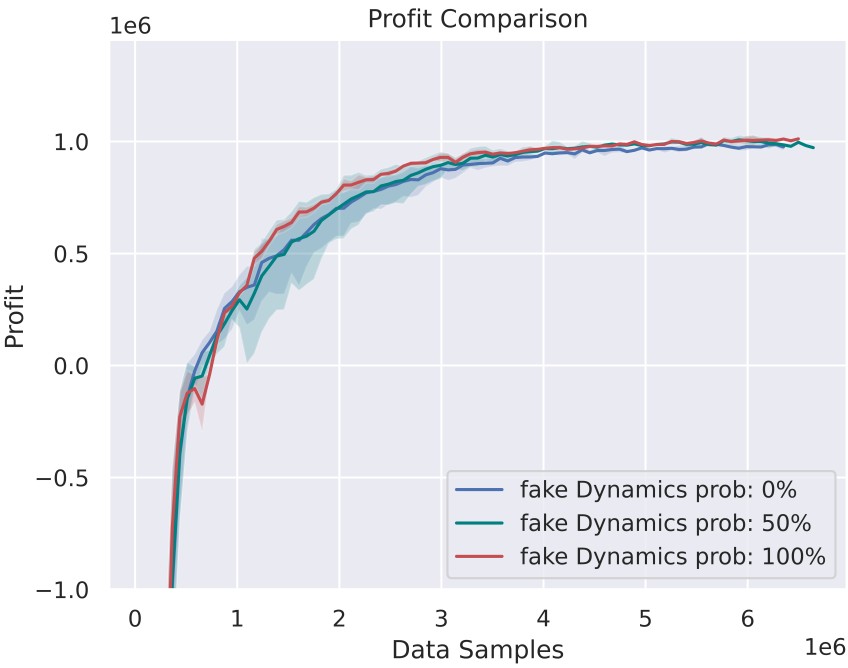

Figure 8: Training curves of CD-PPO with varied ratio of augmented data

**Q2: Does dynamics augmentation improve the performance of the algorithm? If so, how much should we perturb the original data?**

We run similar experiments on environment N5-C100 with CD-PPO, in which the local simulator is ingested with a mixture of original dynamics data and LSTM generated data. The ratio of perturbed dynamics data varies from 0% to 100%. And we found that the algorithm turns out the best performance when we use fully generated data, as shown in Figure 8.

