# OpenReview forum: "Multi-Agent Reinforcement Learning with Shared Resource in Inventory Management"
_ICLR.cc/2022/Conference — ICLR 2022 Submitted_

### Official Review · Reviewer_xoVm · 2021-11-01

**Correctness:** 3
**Technical Novelty And Significance:** 3
**Empirical Novelty And Significance:** 3
**Recommendation:** 6
**Confidence:** 4

**Main Review:**

Strengths:
- The paper tackles an important problem with impact in the real world.
- The formulation as a stochastic game where the SKUs are the players is novel and potentially useful.
- The approximations to make simulations more efficient are sensible.
- Simulations show that the proposed approach is more applicable than the considered baselines.
- The paper is well written and easy to follow.

Weaknesses:
- Although the paper tackles a real world problem, the scenarios used for the experiments are too simple; and although the authors claim that the approach could be used for more complex settings, the fact is that the authors acknowledge scalability issues. Hence, it is difficult to know if this approach will actually work in the real world. I would have expected to see the algorithm working with more realistic settings, including at least hundreds of SKUs, a multi-echelon network and ideally stochastic lead times. The study of the response under disruptions would have been especially appreciated.
- Moreover, the baselines chosen for the experiments aren't very relevant. First, they are not well suited for the problem formulation where there are many SKUs, but they could still work for other formulations of the same inventory management problem. Second, I would have expected comparison with well known operations research baselines.
- The authors also claim their approach could work for non-homogeneous SKUs, but it is not clear how. Would they consider the agent's ids (or features) as inputs to the neural networks?
- Another point for improvement is the comparison with related work. For example, I think the current approach is very related to potential games and to mean-field games. Comparison with the former could bring better theoretical understanding and if the current game is potential (as I suspect), the authors could present their approach as a distributed method and compare with the potential-based centralised approach. Comparison with the latter would be very interesting as they potentially could scale the current formulation to arbitrarily large number of SKUs by relying on a mean-field formulation.

Minor comments
- Some simulation results are not clearly displayed. For example, what is the rationale for the bold font in Table 1? IPPO-IR (w/o context) performs almost as well as CD-PPO and with less variance in the N50-C2000 scenario. Also, why comparing with the worst baseline and not the median or the second best profit in Figure 7?
- Ref to A2C is not Tsitsiklis & Konda, 2000, but an post in the OpenAI blog that suggest a synchronous version of A3C.

**Summary Of The Paper:**

The authors formulate an inventory optimisation problem as a stochastic game, where each SKU is a player. In order to reduce computational complexity, the game is formulated such that the only interactions among players are through the available storage space they have to share. The paper also proposes an algorithm that relies on local simulators (as opposed to joint simulation) and surrogate losses to improve efficiency. Simulations show that the proposed algorithm is able to outperform other MARL baselines for up to 50 SKUs.

**Summary Of The Review:**

The introduces a potentially useful approach for an important real-world problem. But the benefit of tackling a real-world problem vanishes because the proposed approach has been evaluated in very simple environments and with baselines that are not very relevant for this problem. The paper would have higher impact with some of these improvements:
- The algorithm is evaluated in a realistic environment.
- The algorithm is compared with relevant IM baselines (operation research approaches actually used in the real life) even in this toy scenarios.
Also, the paper misses two important lines of related work, namely potential and mean-field games, which could offer new insights to the  proposed approach.


Meta-review
=========
I think the paper tackles an important real world problem with an interesting approach and, although the experiments were a bit lacking, the authors have included a sensible baseline and a more realistic scenario.

I still see some scalability issues and I would appreciate a deeper and insightful discussion with related approaches, like potential and mean-field games. But I think the paper is in better shape than before and it might be useful for other researchers in the field.

So I am happy to increase my score to 6 if the other reviews don't see major flaws.

---

> ### Author Response · Authors · 2021-11-20
> **Response to reviewer xoVm (part 1/2)**
>
> We thank the useful suggestions from the reviewers! We have taken all the comments into consideration and summarized the responses as follows:
>
> > Although the paper tackles a real world problem, the scenarios used for the experiments are too simple; and although the authors claim that the approach could be used for more complex settings, the fact is that the authors acknowledge scalability issues. Hence, it is difficult to know if this approach will actually work in the real world. I would have expected to see the algorithm working with more realistic settings, including at least hundreds of SKUs, a multi-echelon network and ideally stochastic lead times. The study of the response under disruptions would have been especially appreciated.
>
> **Response:**
>
> First, we adopt a public dataset with real demand data.
>
> Second, our algorithm could be applied to more complex scenarios, even though we didn't include these results in the previous version due to time limit. To confirm this, we extend our method to scenarios with stochastic leading time and hundreds of SKUs (VLTs are modelled as exponetial distribution.). The experiment results are listed in Table5 of our revised mannuscript.
>
> Finally, we agree that IM with a multi-echelon network will be more interesting and we definitely will consider such scenarios in our future work. We focus on IM for one store in this work and propose the shared-resource stochastic games to model the shared resource structure in this problem. We think that a scalable algorithm in our simplified scenario is a critical step before we move forward to a more realistical scenario with multi-echelon etc.
>
> ---
>
> > The baselines chosen for the experiments aren't very relevant. First, they are not well suited for the problem formulation where there are many SKUs, but they could still work for other formulations of the same inventory management problem. Second, I would have expected comparison with well known operations research baselines.
>
> **Response:**
>
> 1. We have added a well-known algorithm "base-stock" from OR community as our baseline. Base-stock utilizes Linear Programming to work out the base stock levels and thus the replenishing policy. "Static" indicates that the base stock levels are based on demand data from training set and then kept static while making reorder desicions on test set. "Dynamic" updates its base stock levels on a regular time cycle. "Oracle" directly access the whole test set to calculate its base stock levels, which is essentially a cheating version used to show the upper limits of Base-stock policy.
>
> 2. The goal of our work is to solve the IM problem with massive agents. It is because current MARL algorithms do not work well for such scenarios. The algorithm CD-PPO we proposed in this paper addresses such a problem by leveraging its shared resource structure.
>
> ---
>
> > The authors also claim their approach could work for non-homogeneous SKUs, but it is not clear how. Would they consider the agent's ids (or features) as inputs to the neural networks?
>
> **Response:**
>
> We apologize for being unclear. If we need to deal with non-homogeneous SKUs, we can add agent id as input, or simply use different policy networks for different agents. However, the goal here is to learn policy functions that can generalize, therefore it is more reasonable to consider homogeneous SKUs.
>
> ---
>
> > The current approach is very related to potential games. Comparison with it could bring better theoretical understanding and if the current game is potential (as I suspect), the authors could present their approach as a distributed method and compare with the potential-based centralized approach.
>
> **Response:**
>
> To the best of our knowledge, a game is said to be a potential game if the incentive of all players to change their strategy can be expressed by a single global function called the potential function. It is not clear if there exists a potential function for our problem.
>
> Moreover, if the potential-based centralized approach needs to learn a centralized critic, then it has the same disadvantage as MAPPO. With massive agents, it could be computationally intractable to learn such a centralized critic.

---

> ### Author Response · Authors · 2021-11-20
> **Response to reviewer xoVm (part 2/2)**
>
> > The current approach is very related to Mean-Field Games. Comparison with the latter would be very interesting as they potentially could scale the current formulation to arbitrarily large number of SKUs by relying on a mean-field formulation.
>
> **Response:**
>
> Mention in the reply for Reviewer #yMmz, we describe the differences between our approach and Mf-MARL methods. Nevertheless, MF-MARL has the same disadvantage as MAPPO, which is computationally intractable when there are massive agents.
>
> Due to time limit, we cannot conduct experiment using MF-MARL for the review.  We will try our best to deliver the corresponding results in our next version.
>
> ---
>
> > What is the rationale for the bold font in Table 1?
>
> **Response:**
>
> It is our mistake to write two bold results in the same line of Table1. We will correct it in the revised manuscript.
>
> ---
>
> > IPPO-IR (w/o context) performs almost as well as CD-PPO and with less variance in the N50-C2000 scenario.
>
> **Response:**
>
> We speculate that this is because agents are almost completely independent since the capacity of the shared resource is too large under the N50-C2000 scenario. In other words, the chance of preemption is very small, therefore the context is not useful for value estimation. In IPPO-IR(w/o context), it doesn't use context which avoids the perturbation of context in training. On the contrary, as we mentioned in the paper, in the CD-PPO training process, we use diverse trajectories of context to run local simulations,  which will make agents need to learn to deal with various contexts, i.e. policy in CD-PPO is context-aware. As a result, CD-PPO has a higher variance compared with IPPO-IR(w/o context).
>
> ---
>
> > Why comparing with the worst baseline and not the median or the second best profit in Figure 7?
>
> **Response:**
>
> Thanks for your advice. We will add figures for the median or the second-best profit in our revised manuscript.
>
> ---
>
> > Ref to A2C is not Tsitsiklis & Konda, 2000, but an post in the OpenAI blog that suggest a synchronous version of A3C.
>
> **Response:**
>
> Thanks for your advice. We will add the correct reference.

---

### Official Review · Reviewer_yMmz · 2021-11-02

**Correctness:** 3
**Technical Novelty And Significance:** 2
**Empirical Novelty And Significance:** 2
**Recommendation:** 5
**Confidence:** 4

**Main Review:**

Strengths
- The paper studies a very interesting and promising industrial application of RL. Improvement in inventory replenishment has profound real-world impact.
- A model-based MARL algorithm is proposed to tackle the challenge of large number of agents by exploiting the structure of the multi-SKU replenishment problem, that is, the replenishment of the SKUs are associated only through the global capacity constraint.
- Empirical results appear to support that the proposed method achieves greater learning efficiency than the model-free baselines.

Weaknesses
- It's not clear how exploration is done in the proposed method. In order for the individual SKUs to train and update their policies independently, the global simulator has to run to obtain a sample path of the shared resource state (context) in each epoch. This basically allocates the capacity for each SKU using the previous policy, and each agent only learns within its allocated capacity trajectory. The inner loop of training is very restrictive in this way given that capacity constraint is a core element in inventory replenishment. How does the joint policy explore different capacity allocations?
- The objective function does not incorporate any penalty for inventory overflow, which is a serious issue in practice.
- The surrogate loss function is based on the assumption that each SKU has limited impact to its context (shared resource). When the number of SKUs is small, this would not hold. In the experiments, did you observe any difference in performance over the 5-SKU and 50-SKU cases?
- The paper should add at least one classical non-RL replenishment algorithm as baseline. Otherwise, it is hard to judge the significance of the contribution.
Moreover, mean-field MARL [1] is designed for large number of agents. The idea of the context in this method is similar in principle to MF-MARL.
[1] Yang, Y., Luo, R., Li, M., Zhou, M., Zhang, W. and Wang, J., 2018, July. Mean field multi-agent reinforcement learning. In International Conference on Machine Learning (pp. 5571-5580). PMLR.
- The results compare the number of environment interactions. I assume this is the interaction with the global environment. In the proposed method, each agent further interacts with its local environment for a number of steps, while the model-free methods do not. Hence, the comparison may not be entirely fair.

**Summary Of The Paper:**

This paper considers the multi-period inventory replenishment problem for a single store with multiple SKUs. The problem is formulated as a resource constrained stochastic game, where the store capacity is the resource constraint shared by all SKUs. The paper takes a multi-agent RL approach to solve this problem, with each SKU being modeled as an agent and having its own replenishment policy. To tackle the challenge presented by the potentially large number of SKUs (agents), a context-aware decentralized PPO method is proposed, where each agent is trained independently in its local simulator given a sample path of the shared resource state (context). Benchmarking were performed on two domains with 5 SKUs and 50 SKUs each against several model-free MARL baselines. Results show that the proposed method can achieve comparable results but with much higher sample efficiency.

**Summary Of The Review:**

This paper presents some novelty in addressing the issue of large number of SKUs (agents) in applying MARL to solve the multi-SKU inventory replenishment problem, by exploiting the structure of the problem and devising a model-based approach. The empirical results show that the proposed method achieves better sample efficiency in the global environment than the model-free baselines. However, there are several important algorithmic issues not investigated in this paper, and the important related method in the literature is not compared with.

---

> ### Author Response · Authors · 2021-11-20
> **Response to reviewer yMmz (part 1/2)**
>
> We thank the useful suggestions from the reviewers! We have taken all the comments into consideration and summarized the responses as follows:
>
> > It's not clear how exploration is done in the proposed method. In order for the individual SKUs to train and update their policies independently, the global simulator has to run to obtain a sample path of the shared resource state (context) in each epoch. This basically allocates the capacity for each SKU using the previous policy, and each agent only learns within its allocated capacity trajectory. The inner loop of training is very restrictive in this way given that capacity constraint is a core element in inventory replenishment. How does the joint policy explore different capacity allocations?
>
> **Response:**
>
> We agree that exploration can be hindered by re-using the sampled path of context in local simulators. To address this problem, we additionally perform two data augmentation methods on context trajectories, which help us explore different context allocations.
>
> Besides, our method is on-policy. In the next training iteration, the updated stochastic policy will generate new context trajectories in the global simulator, which can also explore different context allocations since the policy is updated.
>
> ---
>
> > The objective function does not incorporate any penalty for inventory overflow, which is a serious issue in practice.
>
> **Response:**
>
> The penalty for inventory flow is added in the reward function(defined in Eq.6).  We proportionally discard the excessive amount of products as a penalty when facing inventory flow.
>
> ---
>
> > The surrogate loss function is based on the assumption that each SKU has limited impact to its context (shared resource). When the number of SKUs is small, this would not hold. In the experiments, did you observe any difference in performance over the 5-SKU and 50-SKU cases?
>
> **Response:**
>
> First, our goal is to solve IM problem with massive SKUs. The assumption that each SKU has a limited impact on its context (shared resource) holds well for the above scenario.
>
> Second, we agree with the reviewer that our assumption does not hold well when the SKU number is small. In fact, the proposed approach mainly targets for the scenario with large SKU number. On the other hand, under the scenario with small SKU number, many existing algorithms like MAPPO can achieve good results according to our experiments.
>
> Third, we do observe some instability issues of CD-PPO on 5-SKU compared with the 50-SKU case. To deal with the instability issues empirically, we use a smaller learning rate for CD-PPO on the 5-SKU case and find CD-PPO get comparable performance with MAPPO.
>
> ---
>
> > The paper should add at least one classical non-RL replenishment algorithm as baseline. Otherwise, it is hard to judge the significance of the contribution.
>
> **Response:**
>
> We appreciate the comment and have added a well-known algorithm "base-stock" from OR community as our baseline. Below lists all the results.
>
> Base-stock utilizes Linear Programming to work out the base stock levels and thus the replenishing policy. "Static" indicates that the base stock levels are based on demand data from training set and then kept static while making decisions on test set.  "Dynamic" updates its base stock levels on a regular time cycle. "Oracle" directly access the whole test set to calculate its base stock levels, which is essentially a cheating version used to show the upper limits of Base-stock policy.

---

> ### Author Response · Authors · 2021-11-20
> **Response to reviewer yMmz (part 2/2)**
>
> > Mean-field MARL is designed for large number of agents. The idea of the context in this method is similar in principle to MF-MARL.
>
> **Response:**
>
> As pointed in Sec.5, an obvious flaw for CTDE/CTCE is that state-action spaces grow exponentially by the number of agents. Common-used scenarios in MARL frameworks is assumed that the system state can be abstracted and this global state is not relative to the number of agents. By assuming this, the core challenge is to handle the exponential growth in action space. In other words, this makes it infeasible to learn standard individual Q-function $Q^i(s,\boldsymbol{a})$. And to address this issue, MF-MARL[1] is proposed, whose idea is to factorize the Q-function using only the pairwise local interactions and then used mean-field approximator $Q^i(s, a^i, a^{-i})$ to estimate $Q^i(s,\boldsymbol{a})$. In general, there are at least two differences between our approach and MF-MARL methods:
>
> 1. The key difference is that the interaction mechanism is not the same. MF-MARL assumes that agents can interact directly and only grasping the influence of neighbors is enough in the massive-agent setting. However, in IM problem with shared resources, there is no direct interaction between agents — **agents can only exert limited indirect influence through the shared resource****(i.e., context)**. And item "limited" means that from the individual agent's view, the dynamics of the context is an aggregation of all other agents' influences and it cannot be factorized to quantify how a specific agent affects the current agent. So that we believe that MF-MARL is not suitable for the IM problem since its assumption about how agents interact with each other does not hold for inventory management with shared resources.
>
> 2. Due to the abstracted state in Stochastic Games, MF-MARL can scale up to the massive-agents setting since it addresses the exponential growth of action space. However, in the IM problem, the system state is the result of the concatenation of individual states of each agent with extra context. So that MF-MARL will also be suffered from this issue like other centralized MARL methods. And our CD-PPO avoids this issue by estimating the value function only with the individual state of each agent and the context.
>
> 3. Specifically, the way of factorization is also very different. MF-MARL uses pairwise interaction $Q^i(s, a^i, a^{-i})$ to approximate$Q^i(s,\boldsymbol{a})$. On the contrary, CD-PPO uses $Q^i(s^i, a^i, c)$ as its approximation. We need to emphasize that, the core idea of MF-MARL is to use pairwise local interaction of actions as an approximation, while our method uses the context not the input actions to reflect the collective behavior of other agents.
>
> References:
>
> [1] Yang Y, Luo R, Li M, et al. Mean field multi-agent reinforcement learning[C]//International Conference on Machine Learning. PMLR, 2018: 5571-5580.
>
> ---
>
> > The results compare the number of environment interactions. I assume this is the interaction with the global environment. In the proposed method, each agent further interacts with its local environment for a number of steps, while the model-free methods do not. Hence, the comparison may not be entirely fair.
>
> **Response:**
>
> We apologize for being unclear. We compare the number of data samples, from both global and local environments. Specifically, for one interaction in the global environment with N agents, we consider it as N data samples.  For one interaction in the local simulator(based on context trajs) with one agent, we consider it as 1 data sample. We find that comparing by data samples could be the fairest way to compare different approaches.

---

> > ### Author Response · Authors · 2021-11-23
> > **Addendum**
> >
> > In our [response to reviewer xoVm](https://openreview.net/forum?id=-uZp67PZ7p&noteId=432l0fI7j5d), we added additional discussion about MF-MARL framework.

---

### Official Review · Reviewer_Uk4L · 2021-11-03

**Correctness:** 3
**Technical Novelty And Significance:** 3
**Empirical Novelty And Significance:** 2
**Recommendation:** 5
**Confidence:** 4

**Main Review:**

Strengths:
A new MARL algorithm to deal with joint replenishment problem in inventory management is proposed.

Weaknesses:
The claims of the paper about the performance of the algorithm is not well supported with numerical experiments. Also, there are some questions in the paper that need to be addressed. See below comments for more details.

Q1- With the introduced shared contextual information, still, the env is non-stationary from the viewpoint of each agent. In other words, if agent $i$ is in state $s_t^i$ and takes $a_t^i$, the distribution of next state $s_{t+1}^i$ and the reward $r_t^i$ heavily depend on the actions of other agents. How do you deal with that? How can one make sure that the training is not hindered with that? In other words, can you guarantee that the training is successful for any data/demand distribution, any number of SKUs, and other characteristics of the problem?
Q1-1 Same issue exists when you do sampling $c^{-i}^t$ and $(s^i_t; a^i_t; c^i_t)$ from the old policy.


Q2- Since in theory, the algorithms with independent learners may not work in practice, do you have any intuition why does CD-PPO work on the presented examples? Does the data have any given feature/characteristic?
Q2-1 In a broader question, can you categorize the type of problems that CD-PPO work well for them?


Minor comments:
Some of the references do not have all the required information to find the paper, like the journal, doi, etc, like:

Yaodong Yang, Rui Luo, Minne Li, Ming Zhou, Weinan Zhang, and Jun Wang. Mean field multiagent reinforcement learning, 2020. A.1

Georgios Papoudakis, Filippos Christianos, Lukas Sch¨afer, and Stefano V. Albrecht. Benchmarking multi-agent deep reinforcement learning algorithms in cooperative tasks, 2021. 4.1, E.1

Afshin Oroojlooyjadid, MohammadReza Nazari, Lawrence Snyder, and Martin Tak´aˇc. A deep qnetwork for the beer game: A deep reinforcement learning algorithm to solve inventory optimization problems, 2020a. 5.1

Christian Schroeder de Witt, Tarun Gupta, Denys Makoviichuk, Viktor Makoviychuk, Philip H. S. Torr, Mingfei Sun, and Shimon Whiteson. Is independent learning all you need in the starcraft multi-agent challenge?, 2020. 4.1, 5.2

Please take check all the references and update them with the latest version.


[1] Khouja, M., & Goyal, S. (2008). A review of the joint replenishment problem literature: 1989–2005. European Journal of Operational Research, 186(1), 1-16.
[2] Otero-Palencia, C., Amaya–Mier, R., & Yie-Pinedo, R. (2019). A stochastic joint replenishment problem considering transportation and warehouse constraints with gainsharing by Shapley Value allocation. International Journal of Production Research, 57(10), 3036-3059.
[3] Ongkunaruk, P., Wahab, M. I. M., & Chen, Y. (2016). A genetic algorithm for a joint replenishment problem with resource and shipment constraints and defective items. International Journal of Production Economics, 175, 142-152.
[4] Nguyen, T. T., Nguyen, N. D., & Nahavandi, S. (2020). Deep reinforcement learning for multiagent systems: A review of challenges, solutions, and applications. IEEE transactions on cybernetics, 50(9), 3826-3839.
[5] OroojlooyJadid, A., & Hajinezhad, D. (2019). A review of cooperative multi-agent deep reinforcement learning. arXiv preprint arXiv:1908.03963.
[6] Zhang, K., Yang, Z., & Başar, T. (2021). Multi-agent reinforcement learning: A selective overview of theories and algorithms. Handbook of Reinforcement Learning and Control, 321-384.
[7] Minner, S., & Silver, E. A. (2005). Multi-product batch replenishment strategies under stochastic demand and a joint capacity constraint. IIE Transactions, 37(5), 469-479.
[8] Hoque, M. A. (2006). An optimal solution technique for the joint replenishment problem with storage and transport capacities and budget constraints. European journal of operational research, 175(2), 1033-1042.

**Summary Of The Paper:**

A multi-agent RL algorithm is proposed to deal with the inventory management problem where there are several SKUs. Since items should compete for a limited budget and warehouse space, the replenishment problem is modeled as a multi-agent game and a multi-agent reinforcement learning algorithm is proposed to solve that. The agents use a shared piece of information which is the utilized warehouse space and other shared resources (if any) and rather than that there is no other shared information among them. So, each agent uses that shared information along with its local observation, and local reward and train decentralized agents in an independent learner scheme. The PPO algorithm is used to train the independent policies. The agents share the parameters of the actor and critic to mitigate the computation time and power. The action is selecting one of 15 possible values to multiply to the sum of sales over the last two weeks. The algorithm is evaluated on instances with 5 and 50 SKUs and compared with some benchmarks.


**Summary Of The Review:**

Same as above
I'll update my recommendation based on the answers of the authors.

---

> ### Author Response · Authors · 2021-11-20
> **Response to reviewer Uk4L**
>
> Thank you for the thoughtful and constructive suggestions! We have taken all the comments into consideration and summarized the responses as follows:
>
> > With the introduced shared contextual information, still , the env is non-stationary from the viewpoint of each agent.
>
> **Response:**
>
> We apologize for causing confusion about non-stationary here. Please find some clarification about the difference between our methods and existing methods(IPPO, MAPPO) below.
>
> Independent learning methods, such as IPPO, are considered to suffer from the non-stationary problem, while joint action learning methods, such as MAPPO, address the non-stationary problem by introducing a centralized critic. Specifically, IPPO uses $G_t - V(s^i)$ as an advantage estimator to update policy function, while MAPPO uses $G_t - V(s_1, \dots ,s_n)$. We can see that MAPPO has a lower gradient variance than IPPO by using a centralized critic which takes more information as its input.
>
> In contrast, our method uses $G_t - V(s^i, c^i, c^{-i})$ as the advantage estimator to update the policy function. On one hand, our method introduces more information and therefore has a lower gradient variance than IPPO. Since the transition/reward function for each agent largely depends on context, our estimator can reduce gradient variance by a large margin compared with IPPO. On the other hand, our method does not require too much information as in MAPPO, which could be computationally intractable when massive agents are involved.
>
> From the experiment results, we can see that IPPO works well and our method works better than IPPO, while MAPPO cannot scale up to scenarios with more than 50 agents. Although the env is still non-stationary for each agent after introducing context, we did reduce the variance of policy gradient by introducing context as input compared with IPPO.
>
> ---
>
> > How do you deal with that? How can one make sure that the training is not hindered with that?
> In other words, can you guarantee that the training is successful for any data/demand distribution, any number of SKUs, and other characteristics of the problem?
>
> **Response:**
>
> Our method is on-policy. The policy, during the training, will not change much after each single iteration, which makes the training procedure stable. Specifically, the value function for agent $i$ depends on its policy $\pi_i$ and other agents' policies $\pi_{-i}$. Since $\pi_{-i}$ changes little after each iteration, it is not that difficult to estimate the value function for agent $i$.
>
> Different data/demand distribution, different numbers of SKUs, or stochastic VLTs will not be an issue for our algorithm. All different scenarios with the above different parameters can be formulated as shared resource stochastic games, therefore can be solved by our algorithm. This is analogous to developing RL algorithms for MDPs, meaning that any problem that can be formulated by an MDP is able to be solved by DQN/PPO/SAC.
>
> To verify this, we choose stochastic VLTs as the representative extra challenge and did extra experiments where VLTs are modelled as exponetial distribution. The results can be found in Table5 of our revised manuscript, which shows our algorithm can learn to accommodate stochastic VLTs well.
>
> ---
>
> > Since in theory, the algorithms with independent learners may not work in practice, do you have any intuition why does CD-PPO work on the presented examples?
>
> **Response:**
>
> First, IPPO works well in practice. As discussed above, the value function for agent $i$ depends on its policy $\pi_i$ and other agents' policies $\pi_{-i}$. The environment is non-stationary from the viewpoint of each agent $i$, since other agents may change their policies $\pi_{-i}$. However, IPPO is on-policy. For each training iteration, $\pi_{-i}$ changes, therefore it is not that difficult to estimate the value function for agent $i$.
>
> Second, CD-PPO works better than IPPO, since we reduce the variance of policy gradient by introducing more information in the critic.  Since the transition/reward function for each agent yields a high dependency on the context, our estimator can reduce gradient variance by a large margin compared with IPPO.
>
> ---
>
> > Does the data have any given feature/characteristic?  In a broader question, can you categorize the type of problems that CD-PPO work well for them?
>
> **Response:**
>
> Our method is proposed to leverage the shared-resource structure in shared-resource stochastic games. As long as problems can be formulated by a shared-resource stochastic game, our method can work well for them. The key characteristic of a shared-resource stochastic game is that agents only interact indirectly through context, i.e. shared resources.
>
> ---
>
> > Some of the references do not have all the required information to find the paper, like the journal, doi, etc.
>
> **Response:**
>
> We thank the reviewer for pointing out this issue. We will supplement all the missing information in the revised manuscript.

---

> ### Author Response · Authors · 2021-11-29
> **Response to reviewer Uk4L**
>
> Welcome to check out our revisions and responses! If you have any concerns, please do not hesitate to discuss them with us. We appreciate the chance of discussion and are eager to hear from you.

---

### Official Review · Reviewer_XZpy · 2021-11-04

**Correctness:** 3
**Technical Novelty And Significance:** 2
**Empirical Novelty And Significance:** 2
**Recommendation:** 6
**Confidence:** 3

**Main Review:**

Overall, this is a nice paper that addresses a simplified version of a major issue faced by retailers (inventory management and capacity control) as a MARL problem. The approach identifies that in the IM problem - the shared context is only really the resource that needs to be divided between different agents.

The paper has a nice explanation of the problem - though it suffers from weak baselines (other PPO methods), rather than trying out a few different approaches (dual SGD, admm etc). The data is rich in sales/demand - but seems to be a simple simulation for other relevant parameters (VLT, price, cost etc). So that would definitely require some improvement to see if the MARL approach can generalize to those scenarios

The reward given to the agent resembles a shadow price, but has a hard truncation to respect the capacity. This isnt very realistic and could probably be improved a bit. Finally, it seems to be only for one time period? So with arbitrary stochastic VLTs, how do the authors control for the fact that order arrivals can be later then expected and then blow the capacity?

Finally, why wouldn't one just treat every SKU as an instance from some joint stochastic process (with context of course) - and do a dual sgd loop at the batch level. Is the stochastic game formulation adding much more than that?

Overall I weakly recommend accepting the paper as it addresses a novel way to approach the capacity control problem.

**Summary Of The Paper:**

The authors propose a MARL style algorithm to solve the capacity modelling problem for the periodic review inventory management problem.

**Summary Of The Review:**

The paper is a nice MARL approach to the IM Capacity Problem. I like the experiments and the modification of the public data used. The baselines are not ideal and the reward function doesn't resemble a real capacity issue - but it is a good step towards a more general IM solution

---

> ### Author Response · Authors · 2021-11-20
> **Response to reviewer XZpy  (part 1/2)**
>
> Thank you for the thoughtful and constructive suggestions! We have taken all the comments into consideration and summarized the responses below:
>
> > Can MARL methods generalize to more complex scenarios(e.g. with stochastic VLTs)?
>
> Actually, VLT for each SKU in N50 and N100 is stochastic during simulation. The real lead time per each procurement can be longer than specified due to lack of upstream vehicles. (Distribution capacity is also considered in the simulator, though not the focus of this paper). To make VLT easier to model, we increased the number of upstream transporting vehicles to circumvent this uncontrollable stochastic factor, and we did extra experiments where VLTs are modeled as exponential distribution. The results are shown in Table5 in our revised manuscript, where we can observe comparable results as in the paper.
>
> ---
>
> > It seems to be only for one time period? So with arbitrary stochastic VLTs, how do the authors control for the fact that order arrivals can be later than expected and then blow the capacity?
>
> **Response:**
>
> MARL methods can naturally handle stochastic environments. MARL algorithms are developed to solve Stochastic Games(or decentralized POMDP), in which the state transition function is stochastic. As a result, the stochastic VLTs can also be modeled by the stochastic state transition function, which can be solved by MARL methods.
>
> The problem is expected to become harder if we replace deterministic VLTs with stochastic VLTs. We show the experimental results for stochastic VLTs above. It is clear to see that the performance drops a little bit as expected, as each SKU tends to increase the amount of order compared with the deterministic VLTs setting.
>
> If the order arrives later and then blows the capacity, we will proportionally discard the excessive amount of products, which will cause a penalty in the reward function (as defined in Eq.6). This can partially explain why the capacity constraint is designed in the reward function. Ideally, after training we would obtain a policy that can largely avoid such capacity overflows. An alternative way is to treat warehouse capacity as a hard constraint that cannot be broken. In this way, it is necessary to establish a mask mechanism at each step to avoid actions that would cause capacity blows. However, it would become challenging to consider stochastic VLTs.
>
> ---
>
> > The reward given to the agent resembles a shadow price, but has a hard truncation to respect the capacity. This isn't very realistic and could probably be improved a bit.
>
> **Response:**
>
> We admit that the hard truncation in reward function with respect to the capacity is a simplified model of real-world scenarios. The goal is to learn policies that do not exceed the capacity. There are several alternative ways to do so, such as adding constraints on the amount of all products, or adding constraints on the action space when the warehouse is almost full.
>
> In this paper, we choose the simplest way that the excess SKUs will be discarded proportionally according to ρ defined in Eq6, ignoring whether some products have higher prices or can bring larger profits. The advantage of our model is two-fold.
>
> First, the reward/transition function for each product $i$ only depends on $s^i$, $\hat{c}^i$ , $\hat{c}^{-i}$, and $a^i$. In other words, all agents are weakly coupled under this formulation. This weakly coupled structure is critical for developing more efficient MARL algorithms that can leverage such structures. Second, our model is compatible with stochastic VLTs, as discussed above.

---

> ### Author Response · Authors · 2021-11-20
> **Response to reviewer XZpy (part 2/2)**
>
> > The paper has a nice explanation of the problem - though it suffers from weak baselines (other PPO methods), rather than trying out a few different approaches (dual SGD, admm etc). Why wouldn't one just treat every SKU as an instance from some joint stochastic process (with context of course) and do a dual sgd loop at the batch level. Is the stochastic game formulation adding much more than that?
>
> **Response:**
>
> Traditional methods,  which build the joint stochastic process and optimize profits by (dual) SGD and ADMM,  will suffer from the following challenges if being applied to large-scale IM problems.
>
> 1. When there are tens of thousands of variables in the stochastic process, it could be extremely costly to model and optimize the problem. To leverage optimization-based methods, we have to build the stochastic process for all related variables.
>
> 2. We have to solve such an optimization problem for each scenario. We cannot learn a policy generalized to similar yet unseen scenarios.
>
> As a result,  we did not use dual SGD as a baseline in our paper. But we add a classical OR-based method, "base-stock", as the baseline in our new version. Compared with traditional methods, MARL methods have the following advantages.
>
> 1. MARL methods directly learn the dependency between the agent's action and the expected return(future profits) from sampled data produced by a black-box simulator. We can avoid building any complex optimization model of thousands of variables, which is computationally intractable in large-scale problems.
>
> 2. By feature engineering for the state,  the learned policy can generalize to similar yet unseen problems. We do not have to solve the optimization problem for every scenario.

---

> ### Author Response · Authors · 2021-11-29
> **Response to reviewer XZpy**
>
> Welcome to check out our revisions and responses! If you have any concerns, please do not hesitate to discuss them with us. We appreciate the chance of discussion and are eager to hear from you.

---

### Decision · Program_Chairs · 2022-01-20

**Decision:**

Reject

**Comment:**

This is a pretty nice paper, but it suffers a bit from being in an 'uncanny valley' between application and research.  The approach clearly has been made, and derives from, the application under consideration.  However, the application is not a real application, and rather is a simplified simulation. That's okay, but it means that the application here is not the real goal.  So, our attention should go to the solution technique.  Unfortunately, this seems rather specific, exploiting known structure for the specific problem at hand, and lacking other reasonable baselines one could imagine.

So, this is not really an application paper, as the application in the paper is a proxy.  But this is also not really an algorithm paper, as the algorithm is not clearly shown to be generalisable to other settings.  And this is also not a theory paper that tells us something general and meaningful.

These are just observations - this is not criticism per se.  But it means I struggle a little to find something meaningful to learn from this paper, that could be applied elsewhere.

This, in addition to the overall recommendations by the reviewers, unfortunately lead me to reject the paper in its current form.  I want to thank the authors for engaging with the discussion, and hope they have found it interesting and rewarding, despite the outcome this time around.